# Incorporating Unlabelled Data into Bayesian Neural Networks

**Mrinank Sharma**                                                    *mrinank@robots.ox.ac.uk*
*University of Oxford, UK*

**Tom Rainforth**                                                      *rainforth@stats.ox.ac.uk*
*University of Oxford, UK*

**Yee Whye Teh**                                                       *y.w.teh@stats.ox.ac.uk*
*University of Oxford, UK*

**Vincent Fortuin**                                                    *vincent.fortuin@tum.de*
*Helmholtz AI, Munich, Germany*
*Technical University of Munich, Germany*

**Reviewed on OpenReview:** *https://openreview.net/forum?id=q2AbLOwmHm*

## Abstract

Conventional Bayesian Neural Networks (BNNs) are unable to leverage unlabelled data to improve their predictions. To overcome this limitation, we introduce *Self-Supervised Bayesian Neural Networks*, which use unlabelled data to learn models with suitable prior predictive distributions. This is achieved by leveraging contrastive pretraining techniques and optimising a variational lower bound. We then show that the prior predictive distributions of self-supervised BNNs capture problem semantics better than conventional BNN priors. In turn, our approach offers improved predictive performance over conventional BNNs, especially in low-budget regimes.

## 1 Introduction

Bayesian Neural Networks (BNNs) are powerful probabilistic models that combine the flexibility of deep neural networks with the theoretical underpinning of Bayesian methods (Mackay, 1992; Neal, 1995). Indeed, as they place priors over their parameters and perform posterior inference, BNN advocates consider them a principled approach for uncertainty estimation (Wilson & Izmailov, 2020; Abdar et al., 2021), which can be helpful for label-efficient learning (Gal et al., 2017). It has even recently been argued that improving them will be crucial for large language models (Papamarkou et al., 2024) and generative AI as a whole (Manduchi et al., 2024).

Conventionally, BNN researchers have focused on improving predictive performance using human-crafted priors over network parameters or predictive functions (e.g., Louizos et al., 2017; Tran et al., 2020; Matsubara et al., 2021; Fortuin et al., 2021a). However, several concerns have been raised with BNN priors (Wenzel et al., 2020; Noci et al., 2021). It also stands to reason that the vast store of semantic information contained in unlabelled data should be incorporated into BNN priors, and that the potential benefit of doing so likely exceeds the benefit of designing better, but ultimately human-specified, priors over parameters or functions. Unfortunately, as standard BNNs are explicitly only models for supervised prediction, they cannot leverage such semantic information from unlabelled data by conditioning on it.

To overcome this shortcoming, **we introduce *Self-Supervised Bayesian Neural Networks*** (§3), which use unlabelled data to learn improved priors over functions. In other words, our approach improves the BNN prior predictive distribution (which we will just call *prior predictive* in the remainder of the paper) by incorporating unlabelled data into it. This contrasts with designing different but ultimately *human-specified* priors, which is the prevalent approach.

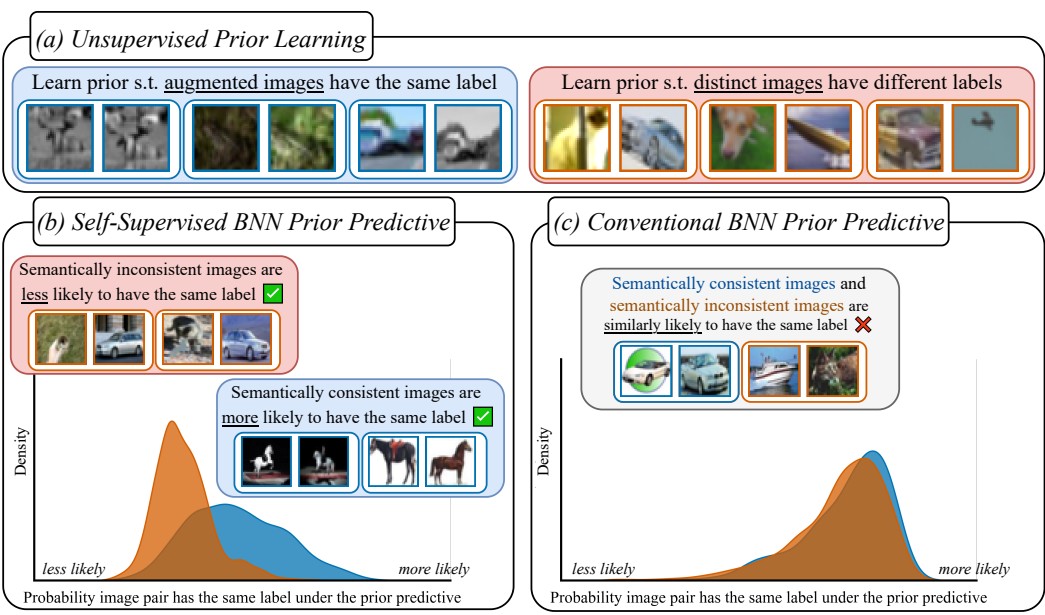

Figure 1: **Self-Supervised Bayesian Neural Networks**. (a) Pre-training in self-supervised BNNs corresponds to unsupervised prior learning. We learn a model with a prior distribution such that augmented images likely have the same label and distinct images likely have different labels under the prior predictive. (b) Self-supervised BNN priors assign higher probabilities to semantically consistent image pairs having the same label compared to semantically inconsistent image pairs. Here, semantically consistent image pairs have the same ground-truth label, and semantically inconsistent image pairs have different ground-truth labels. The plot shows a kernel density estimate of the log-probability that same-class and different-class image pairs are assigned the same label under the prior. (c) Unlike self-supervised prior predictives, conventional BNN prior predictives assign similar probabilities to semantically consistent and semantically inconsistent image pairs having the same label.

In practice, self-supervised BNNs generate pseudo-labelled data using unlabelled data and data augmentation, similar to contrastive learning (Oord et al., 2019; Chen et al., 2020a;b; Grill et al., 2020; Hénaff et al., 2020). We use this generated data to learn models with powerful prior predictive distributions. To do this, we perform unsupervised model learning by optimising a lower bound of a log-marginal likelihood dependent on the pseudo-labelled data. This biases the prior towards functions that assign augmented image pairs a larger likelihood of having the same label than distinct images (Fig. 1a). Following pretraining, we perform inference in the learnt model to make predictions.

**We then further demonstrate that self-supervised BNN prior predictives reflect input-pair semantic similarity better than normal BNN priors** (§4). To do so, we develop a methodology to better understand the prior predictive distributions of BNNs. Our approach is to measure the probability of *pairs* of data points having the same label under the prior. Intuitively, pairs of points that are more semantically similar should be more likely to have the same label under the prior predictive. Applying this methodology, we see that the functional priors learned by self-supervised BNNs distinguish same-class input pairs and different-class input pairs much better than conventional BNNs (Fig. 1b).

Finally, we empirically demonstrate that the improved prior predictives of self-supervised BNNs translate to improved predictive performance, especially in problem settings with few labelled examples (§5).

## 2 Background: Bayesian Neural Networks

Let $f_\theta(x)$ be a neural network with parameters $\theta$ and $\mathcal{D} = \{(x_i, y_i)\}_{i=1}^N$ be an dataset. We want to predict $y$ from $x$. A BNN specifies a prior over parameters, $p(\theta)$, and a likelihood, $p(y|f_\theta(x))$, which in turn define the posterior $p(\theta|\mathcal{D}) \propto p(\theta) \prod_i p(y_i|f_\theta(x_i))$. To make predictions, we approximate the posterior predictive $p(y_\star|x_\star, \mathcal{D}) = \mathbb{E}_{p(\theta|\mathcal{D})}[p(y_\star|f_\theta(x_\star))]$.

Improving BNN priors has been a long-standing goal for the community, primarily through improved human-designed priors. One approach is to improve the prior over the network's parameters (Louizos et al., 2017; Nalisnick, 2018). Others place priors directly over predictive functions (Flam-Shepherd et al., 2017; Sun et al., 2019; Matsubara et al., 2021; Nalisnick et al., 2021; Raj et al., 2023). Both approaches, however, present challenges—the mapping between the network's parameters and predictive functions is complex, while directly specifying our beliefs over predictive functions is itself a highly challenging task. For these reasons, as well as convenience, isotropic Gaussian priors over network parameters remain the most common choice (Fortuin, 2022), despite concerns (Wenzel et al., 2020). In contrast to these works, we propose to *learn* better functional priors from unlabelled data via contrastive learning.

## 3 Self-Supervised BNNs

Conventional BNNs are unable to use unlabelled data to improve their predictions. To overcome this limitation, we introduce *Self-Supervised BNNs*. At a high level, self-supervised BNNs allow unlabelled data to be incorporated by using it to learn a powerful prior that captures known similarities between inputs. In practice, we can utilise ideas from contrastive learning to learn models with prior predictive distributions that reflect the semantics of different input pairs. The high-level idea is thus to use prior knowledge in the form of data augmentations, for which we believe that the semantic content of the data should be invariant to them. We can then use a variational method to learn a function-space prior that assigns more weight to functions, whose outputs on unlabelled data are also invariant to these augmentations.

**Problem Specification.** Suppose $\mathcal{D}^u = \{x_i^u\}_{i=1}^N$ is an unlabelled dataset of examples $x_i^u \in \mathbb{R}^n$. Let $\mathcal{D}^t = \{(x_i^t, y_i^t)\}_{i=1}^T$ be a labelled dataset corresponding to a supervised "downstream" task, where $y_i^t$ is the target associated with $x_i^t$. We want to use both $\mathcal{D}^u$ and $\mathcal{D}^t$ to train a deep learning model for predicting $y$ given $x$ with probabilistic parameters $\theta$, where all information about the data is incorporated through the distribution on $\theta$. That is, we predict using $p(y|x, \theta)$ for a given $\theta$.

### 3.1 Incorporating Unlabelled Data into BNNs

The simplest way one might proceed, is to place a prior on $\theta$ and then condition on both $\mathcal{D}^u$ and $\mathcal{D}^t$, leading to a posterior $p(\theta|\mathcal{D}^u, \mathcal{D}^t) \propto p(\theta|\mathcal{D}^u)\,p(\mathcal{D}^t|\mathcal{D}^u, \theta)$. However, if we are working with conventional BNNs, which are explicitly models for supervised prediction, then $p(\theta|\mathcal{D}^u) = p(\theta)$. Further, as the predictions depend only on the parameters, $p(\mathcal{D}^t|\mathcal{D}^u, \theta) = p(\mathcal{D}^t|\theta)$, which then means that $p(\theta|\mathcal{D}^u, \mathcal{D}^t) = p(\theta|\mathcal{D}^t)$. Thus, we cannot incorporate $\mathcal{D}^u$ by naïvely conditioning on it.

To get around this problem, we propose to instead use $\mathcal{D}^u$ to generate hypothetical labelled data and then condition our predictive model on it. In other words, we will use $\mathcal{D}^u$ to guide a *self-supervised* training of the model, thereby incorporating the desired information from our unlabelled data. To do this, we will draw on data augmentation (Yaeger et al., 1996; Krizhevsky et al., 2012; Shorten & Khoshgoftaar, 2019) and contrastive learning (Oord et al., 2019; Chen et al., 2020b;a; Grill et al., 2020; Hénaff et al., 2020; Chen & He, 2020; Foster et al., 2020; Miao et al., 2023).

Indeed, such approaches that use data augmentation provide effective means for making use of prior information. Although it is difficult to encode our prior beliefs with hand-crafted priors over neural network parameters, we can construct augmentation schemes that we expect to preserve the semantic properties of different inputs. We thus also expect these augmentation schemes to preserve the unknown downstream labels of different inputs. The challenge is now to transfer the beliefs—implicitly defined through our data augmentation scheme—into our model.

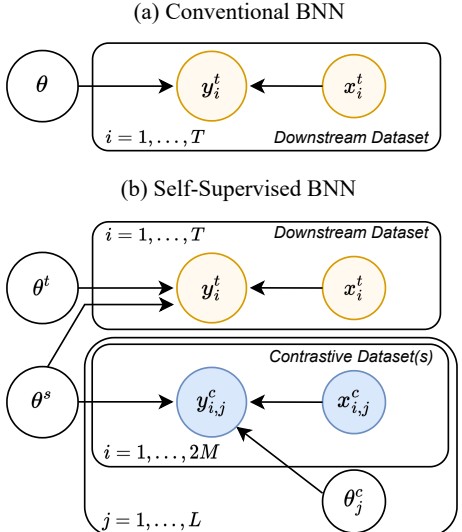

Figure 2: **BNN Probabilistic Models.** (a) Probabilistic model for conventional BNNs. (b) Probabilistic model for self-supervised BNNs. We share parameters between different tasks, which allows us to condition on generated self-supervised data. $j$ indexes self-supervised tasks, $i$ indexes datapoints.

One simple way to do this would be to just augment the data in $\mathcal{D}^t$ when training a standard BNN with standard data augmentation. However, this would ignore the rich information available in $\mathcal{D}^u$. Instead, we will use a construct from contrastive learning to generate pseudo-labelled data from $\mathcal{D}^u$, and then condition $\theta$ on both this pseudo data and $\mathcal{D}^t$.

Concretely, suppose we have a set of data augmentations $\mathcal{A} = \{a : \mathbb{R}^n \to \mathbb{R}^n\}$ that preserve semantic content. We use $\mathcal{A}$ and $\mathcal{D}^u$ to generate a *contrastive dataset* $\mathcal{D}^c$ that reflects our subjective beliefs by:

1. Drawing $M$ examples from $\mathcal{D}^u$ at random, $\{\hat{x}_i\}_{i=1}^M$, where $i$ indexes the subset, not $\mathcal{D}^u$;
2. For each $\hat{x}_i$, sampling $a^A, a^B \sim \mathcal{A}$ and augmenting, giving $\tilde{x}_i^A = a^A(\hat{x}_i)$ and $\tilde{x}_i^B = a^B(\hat{x}_i)$;
3. Forming $\mathcal{D}^c$ by assigning $\tilde{x}_i^A$ and $\tilde{x}_i^B$ the same class label, which is the subset index $i$.

We thus have $\mathcal{D}^c = \{(x_i^c, y_i^c)\}_{i=1}^{2M} = \{(\tilde{x}_i^A, i)\}_{i=1}^M \cup \{(\tilde{x}_i^B, i)\}_{i=1}^M$, where the labels are between 1 and $M$. The task associated with our generated data is thus to predict the subset index corresponding to each augmented example. We can repeat this process $L$ times and create a set of contrastive task datasets, $\{\mathcal{D}_j^c\}_{j=1}^L$. Here, we consider the number of generated datasets $L$ to be a fixed, finite hyper-parameter, but we discuss the implications of setting $L = \infty$ in Appendix A. Note that rather than using a hand-crafted prior to capture our semantic beliefs, we have instead used data augmentation in combination with unlabelled data.

Next, to link each $\mathcal{D}_j^c$ with the downstream predictions, we use parameter sharing (see Fig. 2). Specifically, we introduce parameters $\theta_j^c$ for each $\mathcal{D}_j^c$, parameters $\theta^t$ for $\mathcal{D}^t$, and shared-parameters $\theta^s$ that are used for both the downstream and the contrastive tasks. $\mathcal{D}^u$ thus informs downstream predictions through $\theta^s$, via $\{\mathcal{D}_j^c\}_{j=1}^L$. For example, $\theta^t$ and $\theta_j^c$ could be the parameters of the last layer of a neural network, while $\theta^s$ could be the shared parameters of earlier layers.

**Learning in this Framework.** We now discuss different options for learning in this framework. Using the Bayesian approach, one would place priors over $\theta^s$, $\theta^t$, and each $\theta_j^c$. This then defines a posterior distribution given the observed data $\{\mathcal{D}_j^c\}_{j=1}^L$ and $\mathcal{D}^t$. To make predictions on the downstream task, which depend on $\theta^s$ and $\theta^t$ only, we would then use the posterior predictive:

$$p(y_\star^t | x_\star, \{\mathcal{D}_j^c\}_{j=1}^L, \mathcal{D}^t) = \mathbb{E}_{p(\theta^s | \{\mathcal{D}_j^c\}_{j=1}^L, \mathcal{D}^t)}[\mathbb{E}_{p(\theta^t | \theta^s, \mathcal{D}^t)}[p(y_\star^t | x_\star, \theta^s, \theta^t)]], \quad (1)$$

where we have (i) noted that the downstream task parameters $\theta^t$ are independent of $\{\mathcal{D}_j^c\}_{j=1}^L$ given the shared parameters $\theta^s$ and (ii) integrated over each $\theta_j^c$ and $\theta^t$ in the definition of $p(\theta^s|\{\mathcal{D}_j^c\}_{j=1}^L, \mathcal{D}^t)$.

Alternatively, one can learn a point estimate for $\theta^s$, e.g., with MAP estimation, and perform full posterior inference for $\theta^t$ and $\theta_j^c$ only. This would be a *partially stochastic* network, which Sharma et al. (2023) showed often outperforms fully stochastic networks while being more practical. In the case where all parameters are shared up to the last (linear) layer, this is also known as the *neural linear model* (Lázaro-Gredilla & Figueiras-Vidal, 2010), which has been shown to have many desirable properties (Ober & Rasmussen, 2019; Harrison et al., 2023). Learning in this way is also known as *model learning*, as used in deep kernels and variational autoencoders (Kingma & Welling, 2013; Rezende et al., 2014; Wilson et al., 2015; 2016), where in the case of Gaussian processes (GPs), a point estimate of kernel parameters is learned to also define a learned prior over functions. Note that our approach can also be considered kernel learning, where the representations of input data after the shared layers $\theta^s$ define the reproducing kernel Hilbert space (RKHS) and the kernel is given by their inner products. In learning a point estimate for $\theta^s$, one is learning a suitable model to perform inference in. This model has the following prior predictive:

$$p(y_\star^t|x_\star, \{\mathcal{D}_j^c\}_{j=1}^L) = \mathbb{E}_{p(\theta^t)}[p(y_\star^t|x_\star, \theta_\star^s, \theta^t)], \tag{2}$$

where $\theta_\star^s$ is the learnt value of $\theta^s$.[1] We would then update our beliefs over $\theta^t$ in light of observed data. Through $\theta_\star^s$, we are using $\mathcal{D}^u$ to effectively learn a prior over the functions that can be represented by the network in combination with $\theta^t$. Learning a point estimate for $\theta^s$ is thus our main approach.

## 3.2 Self-Supervised BNNs in Practice

We now use our framework to propose a practical two-step algorithm for self-supervised BNNs.

**Preliminaries.** We focus on image-classification problems. We use an encoder $f_{\theta^s}(\cdot)$ that maps images to representations and is shared across the contrastive tasks and the downstream task. The shared parameters $\theta^s$ thus are the base encoder's parameters. We also normalise the representations produced by this encoder. For the downstream dataset, we use a linear readout layer from the encoder representations, i.e., we have $\theta^t = \{W^t, b^t\}$ and $y_i^t \sim \text{softmax}(W^t f_{\theta^s}(x_i) + b^t)$. The rows of $W_j^t$ are thus class template vectors, that is, a data point will achieve the highest possible softmax probability for a certain class if its representation is equal to (a scaled version of) the corresponding row of the weight matrix. For the contrastive tasks, we use a linear layer without biases, i.e., $\theta_j^c = W_j^c$, and $j$ indexes contrastive tasks. We place Gaussian priors over $\theta^s$, $\theta^t$, and each $\theta_j^c$.

**Pre-training $\theta^s$ (Step I).** Here, we learn a point estimate for the base encoder parameters $\theta^s$, which induces a functional prior over the downstream task labels (see Eq. 2). To learn $\theta^s$, we want to optimise the (potentially penalised) log-likelihood $\log p(\{\mathcal{D}_j^c\}_{j=1}^L, \mathcal{D}^t|\theta^s)$, but this would require integrating over $\theta^t$ and each $\theta^c$. Instead, we use the evidence lower bound (ELBO):

$$\tilde{\mathcal{L}}_j^c(\theta^s) = \mathbb{E}_{q(\theta_j^c)}[\log p(\mathcal{D}_j^c|\theta^s, \theta_j^c)] - D_{\text{KL}}(q(\theta_j^c)||p(\theta_j^c)) \leq \log p(\mathcal{D}_j^c|\theta^s), \tag{3}$$

where $q(\theta_j^c)$ is a variational distribution over the contrastive task parameters.

Rather than learning a different variational distribution for each contrastive task $j$, we amortise the inference and exploit the structure of the contrastive task. The contrastive task is to predict the corresponding source image index for each augmented image. That is, for the first pair of augmented images in a given contrastive task dataset, we want to predict class "1", for the second pair, we want to predict class "2", and so forth. The label is the index within the contrastive dataset, not the full dataset. To predict these labels, we use a linear layer applied to an encoder that produces normalised representations. We want a suitable variational distribution for this linear layer.

To make progress, we define $\tilde{z}_i^A = f_{\theta^s}(\tilde{x}_i^A)$ and $\tilde{z}_i^B = f_{\theta^s}(\tilde{x}_i^B)$, which are the *representations* of given images. To solve the contrastive task well, we want to map $z_1^A$ and $z_1^B$ to class "1", $z_2^A$ and $z_2^B$ to class "2", and so forth.

---

[1] Alternatively, this approach can be understood as learning the prior $p(\theta^s, \theta^t) = p(\theta^t)\delta_{\theta^s=\theta_\star^s}$, where $\delta$ is the Dirac delta function.

---

**Algorithm 1** Self-Supervised BNNs

---

**Input:** augmentations $\mathcal{A}$, unlabelled data $\mathcal{D}^u$, task data $\mathcal{D}^t$, contrastive prior $p(W^c)$
**for** $j = 1, \ldots, L$ **do**                                                                          ▷ Unsupervised prior learning
  Draw subset $\{\hat{x}_i\}_{i=1}^M$, set $\mathcal{D}_j^c = \{\}$
  **for** $i = 1, \ldots, M$ **do**                                                                          ▷ Create contrastive task
    Sample $a^A, a^B \sim \mathcal{A}$
    $\tilde{x}_i^A = a^A(\hat{x}_i)$, $\tilde{x}_i^B = a^B(\hat{x}_i)$
    $\tilde{z}_i^A = f_{\theta^s}(\tilde{x}_i^A)$, $\tilde{z}_i^B = f_{\theta^s}(\tilde{x}_i^B)$
    $\omega_i = 0.5(\tilde{z}_i^A + \tilde{z}_i^B)$
    Add $(\tilde{x}_i^A, i)$ and $(\tilde{x}_i^B, i)$ to $\mathcal{D}_j^c$.
  **end for**
  $W_j^c = \left[\omega_1^T \quad \ldots \quad \omega_M^T\right]/\tau + \epsilon$, with $\epsilon \sim \mathcal{N}(0, \sigma^2 I)$
  $\tilde{\mathcal{L}}(\tau, \sigma^2, \theta^s) = \log p(\theta^s) + \frac{1}{2M}\mathbb{E}_{q(W_j^c)}[p(\mathcal{D}_j^c|\theta^s, W_j^c)] - \bar{D}_{\mathrm{KL}}[q(W_j^c)||p(W^c))]$
  Update $\theta^s, \tau, \sigma^2$ to maximise $\tilde{\mathcal{L}}(\tau, \sigma^2, \theta^s)$
**end for**
Approximate $p(\theta^t|\mathcal{D}^t, \theta^s) \simeq q(\theta^t)$                                                                          ▷ Evaluation
Predict using $\mathbb{E}_{q(\theta^t)}[p(y_\star^t|x_\star, \theta^s, \theta^t)]$

---

We define $\omega_i = 0.5(\tilde{z}_i^A + \tilde{z}_i^B)$, i.e., $\omega_i$ is the mean representation for each augmented pair of images. Because the rows of the linear layer weight matrix $W_j^c$ are effectively class templates, we use $q(W_j^c; \tau, \sigma^2) = \mathcal{N}(\mu_j^c, \sigma^2 I)$ with $\mu_j^c = \left[\omega_1^T \quad \ldots \quad \omega_M^T\right]/\tau$. In words, the mean representation of each augmented image pair is the class template for each source image, which should solve the contrastive task well. Note that since this makes the last-layer weights data-dependent, it also renders them softmax outputs invariant to the arbitrary ordering of the data points in the batch, since if one permutes the $x_i$, and thus $z_i$, this also automatically permutes the $\omega_i$ and thus rows of $W_j^c$ in the same way. Also, recall that the $W_j^c$ are auxiliary parameters that are just needed during the contrastive learning to learn a $\theta^s$ that induces a good functional prior, but are discarded afterwards and not used for the actual supervised task of interest. The variational parameters $\tau$ and $\sigma^2$ determine the magnitude of the linear layer and the per-parameter variance, vary throughout training, are shared across contrastive tasks $j$, and are learnt by maximising Eq. (3) with reparameterisation gradients (Price, 1958; Kingma & Welling, 2013).

Both the contrastive tasks and downstream task provide information about the base encoder parameters $\theta^s$. One option would be to learn the base encoder parameters $\theta^s$ using only data derived from $\mathcal{D}^u$ (Eq. 3), which would correspond to a standard self-supervised learning setup. In this case, the learnt prior would be task-agnostic. An alternative approach is to use both $\mathcal{D}^t$ and $\mathcal{D}^u$ to learn $\theta^s$, which corresponds to a semi-supervised setup. To do so, we can use the ELBO for the downstream data:

$$\tilde{\mathcal{L}}^t(\theta^s) = \mathbb{E}_{q(\theta^t)}[\log p(\mathcal{D}^t|\theta^t, \theta^s)] - D_{\mathrm{KL}}[q(\theta^t)||p(\theta^t)] \leq \log p(\mathcal{D}^t|\theta^s), \qquad (4)$$

where $q(\theta^t) = \mathcal{N}(\theta^t; \mu^t, \Sigma^t)$ is a variational distribution over the downstream task parameters; $\Sigma^t$ is diagonal. We can then maximise $\sum_j \tilde{\mathcal{L}}_j^c(\theta^s) + \alpha \cdot \tilde{\mathcal{L}}^t(\theta^s)$, where $\alpha$ is a hyper-parameter that controls the weighting between the downstream task and contrastive task datasets. We consider both variants of our approach, using Self-Supervised BNNs to refer to the variant that pre-trains only with $\{\mathcal{D}_j^c\}_{j=1}^L$ and Self-Supervised BNNs* to refer to the variant that uses both $\{\mathcal{D}_j^c\}_{j=1}^L$ and $\mathcal{D}^t$ .

**Downstream Inference (Step II).** Having learnt a point estimate for $\theta^s$, we can use any approximate inference algorithm to infer $\theta^t$. Here, we use a post-hoc Laplace approximation (Daxberger et al., 2021).

Algorithm 1 summarises Self-Supervised BNNs, which learn $\theta^s$ with $\mathcal{D}^u$ only. We found tempering with the mean-per-parameter KL divergence, $\bar{D}_{\mathrm{KL}}$, improved performance, in line with other work (e.g., Krishnan et al., 2022). Moreover, we generate a new $\mathcal{D}_j^c$ per gradient step so $L$ corresponds to the number of gradient steps. As shown on Algorithm 1, our full loss is $\tilde{\mathcal{L}}(\tau, \sigma^2, \theta^s) = \log p(\theta^s) + \frac{1}{2M}\mathbb{E}_{q(W_j^c)}[p(\mathcal{D}_j^c|\theta^s, W_j^c)] - \bar{D}_{\mathrm{KL}}[q(W_j^c)||p(W^c))]$.

The first term of this loss is a prior over the shared parameters, in our case a Gaussian prior, which is equivalent to weight decay. The second term is where the actual contrastive learning happens, namely it is an expected Categorical log-likelihood (i.e., cross-entropy) over the softmax logits under $W_j^c$. Recall that the rows of this weight matrix are the mean embeddings vectors $\omega_i$, so this likelihood encourages the inner products $\omega_i^\top \tilde{z}_j^\cdot$ to be large for $i = j$, that is, drawing two augmentations of the same image towards their mean and thus each other, and to be small for $i \neq j$, that is, pushing augmentations of different images away from each other. Finally, the third KL term places a Gaussian prior on $W^c$, which in our case means that the $\omega_i$ that make up this matrix (and thus the embeddings $\tilde{z}_j^\cdot$) cannot grow without bounds to maximize the likelihood score, but have to stay reasonably close together. Moreover, following best-practice for contrastive learning (Chen et al., 2020a), we use a non-linear *projection head* $g_\psi(\cdot)$ *only* for the contrastive tasks. For further details, see Appendix A.

**Pre-training as Prior Learning.**  In this work, our central aim is to incorporate unlabelled data into BNNs. To achieve this, in practice, we perform model learning using contrastive datasets generated from the unlabelled data and data augmentation. This corresponds to an unsupervised prior learning step. Since our objective function during this is a principled lower bound on the log-marginal likelihood, it is similar to type-II maximum likelihood (ML), which is often used to learn parameters for deep kernels (Wilson et al., 2015) of Gaussian processes (Williams & Rasmussen, 2006), and recently also for BNNs (Immer et al., 2021a). As such, similar to type-II ML, our approach can be understood as a form of prior learning. Although we learn only a point-estimate for $\theta^s$, this fixed value induces a prior distribution over predictive functions through the task-specific prior $p(\theta^t)$. However, while normal type-II ML learns this prior using the observed data itself, our approach maximises a marginal likelihood derived from unsupervised data.

## 4   How Good Are Self-Supervised BNN Prior Predictives?

We showed our approach incorporates unlabelled data into the downstream task prior predictive distribution (Eq. 2). We also argued that, as the generated contrastive data encodes our beliefs about the semantic similarity of different image pairs, incorporating the unlabelled data should improve the functional prior. We now examine whether this is indeed the case.

Unfortunately, prior predictive checks are hard to apply to BNNs because of the high dimensionality of the input space. We will therefore introduce our own novel metric to assess the suitability of the prior predictive.

The basis for our approach is to note that, intuitively, a suitable prior should reflect a belief that *the higher the semantic similarity between pairs of inputs, the more likely these inputs are to have the same label.* Therefore, rather than inspecting the prior predictive at single points in input space, we examine the *joint* prior predictive of *pairs* of inputs with known semantic relationships. Indeed, it is far easier to reason about the relationship between examples than to reason about distributions over high-dimensional functions.

Note that this is of course only a reasonable assumption in cases where we believe to have sufficiently good knowledge of semantic similarity in our data domain. That is, we need to have a set of data augmentations for the contrastive tasks, for which we can be reasonably certain that the true labels in our downstream task will be invariant to them. Recent results in contrastive learning suggest that this is indeed the case for natural images paired with the augmentations used in SimCLR (Chen et al., 2020a), which is why we use these in our experiments.

To compute our proposed metric, we consider different groups of input pairs. Each group is comprised of input pairs with known semantic similarity. For example, for image data, we could use images of the same class as a group with high semantic similarity, and image pairs from different classes as a group with lower semantic similarity. To investigate the properties of the prior, we can evaluate the probability that input pairs from different groups are assigned the same label under the prior predictive. We can qualitatively investigate the behaviour of this probability across and within different groups. For a prior to be more adapted to the task than an uninformative one, input pairs from groups with higher semantic similarity should be more likely to have the same label under the prior predictive.

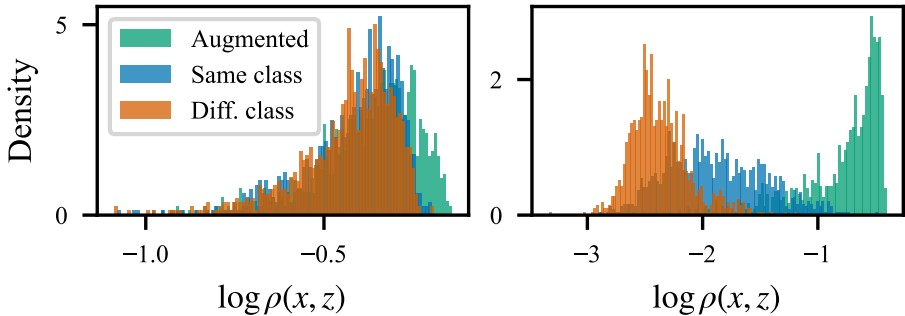

Figure 3: **BNN Prior Predictives.** We investigate prior predictives by computing the probability $\rho$ that particular image pairs have the same label under the prior, and examining the distribution of $\rho$ across different sets of image pairs. We consider three sets of differing semantic similarity: (i) augmented images; (ii) images of the same class; and (iii) images of different classes. Left: Conventional BNN prior. Right: Self-supervised BNN learnt prior predictive. The self-supervised learnt prior reflects the semantic similarity of the different image pairs better than the BNN prior, which is reflected in the spread between the different distributions.

Table 1: **Prior Evaluation Scores**. Mean and standard deviation across three seeds shown. Self-supervised priors are better than standard BNN priors.

| Prior Predictive | Prior Evaluation Score $\alpha$ |
|---|---|
| BNN — Gaussian | $0.261 \pm 0.024$ |
| BNN — Laplace | $0.269 \pm 0.007$ |
| Self-Supervised BNN | $\mathbf{0.680} \pm 0.063$ |

Moreover, we can extend this methodology to quantitatively evaluate the prior. Suppose we have $G$ groups of input pairs, $\mathcal{G}_g = \{(x_i^g, \hat{x}_i^g)_{i=1}^{|\mathcal{G}_g|}\}$ with $g = 1, \ldots, G$, and suppose $\mathcal{G}_1$ is the group with the highest semantic similarity, $\mathcal{G}_2$ is the group with the second highest semantic similarity, and so forth. We define $\rho(x, \hat{x})$ as the probability that inputs $x, \hat{x}$ have the same label under the prior predictive, i.e., $\rho(x, \hat{x}) = \mathbb{E}_\theta[p(y(x) = y(\hat{x})|\theta)]$ where $y(x)$ is the label corresponding to input $x$. We then define *the prior evaluation score*, $\alpha$, as:

$$\alpha = \mathbb{E}[\mathbb{I}(\rho(x^1, \hat{x}^1) > \ldots > \rho(x^G, \hat{x}^G))], \tag{5}$$

where we compute the expectation sampling $(x^1, \hat{x}^1) \sim \mathcal{G}_1$ and so forth. This is the probability that the prior ranks randomly sampled input pairs correctly, in terms of semantically similar groups being assigned higher probabilities of their input pairs having the same label. We now use this methodology to compare conventional BNNs and self-supervised BNNs.

**Experiment Details.** We investigate the different priors on CIFAR10. For the BNN, we follow Izmailov et al. (2021b) and use a ResNet-20-FRN with a $\mathcal{N}(0, 1/5)$ prior over the parameters. For the self-supervised BNN, we learn a base encoder of the same architecture with $\mathcal{D}^u$ only and sample from the prior predictive using Eq. (2). $\theta^t$ are the parameters of the linear readout layer. For the image pair groups, we use: (i) an image from the validation set (the "base image") and an augmented version of the same image; (ii) a base image and another image of the same class; and (iii) a base image and an image of a different class. As these image pair groups have decreasing semantic similarity, we want the first group to be the most likely to have the same label, and the last group to be the least likely. See Appendix B.3 for more details.

**Graphical Evaluation.** First, we visualise the BNN and self-supervised BNN prior predictive (Fig. 1 and 3). The standard BNN prior predictive reflects a belief that all three image pair groups are similarly likely to have the same label, and thus does not capture semantic information well. In contrast, the self-supervised prior reflects a belief that image pairs with higher semantic similarity are more likely to have the same label.

In particular, the self-supervised prior is able to distinguish between image pairs of the same class and of different classes, *even without access to any ground-truth labels.*

**Quantitative Evaluation.** We now quantify how well different prior predictives reflect data semantics. In Table 1, we see that conventional BNN priors reflect semantic similarity much less than self-supervised BNN priors, matching our qualitative evaluation. Note that this measure has of course been designed by us to capture the kind of property in the prior that our contrastive training is meant to induce, and should therefore just be seen as a confirmation that our proposed approach works as expected. There are, naturally, many other properties that one could desire in a prior, which are not captured by this metric.

## 5 Self-Supervised BNNs Excel in Low-Label Regimes

In the previous section, we showed that self-supervised BNN prior predictives reflect semantic similarity of input pairs better than conventional BNNs (§4). One hopes that this translates to improved predictive performance, particularly when conditioning on small numbers of labels, which is where the prior has the largest effect (Gelman et al., 1995; Murphy, 2012). We now show that this is indeed the case. Self-supervised BNNs offer improved predictive performance over standard BNNs, with especially large gains when making predictions given small numbers of observed labels.

### 5.1 Semi-Supervised Learning

**Training Datasets.** We evaluate the performance of different BNNs on the CIFAR10 and CIFAR100 datasets, which are standard benchmarks within the BNN community. We evaluate the performance of different baselines when conditioning on 50, 500, 5000, and 50000 labels from the training set.

**Algorithms.** As baselines, we consider the following BNNs: MAP, SWAG (Maddox et al., 2019), a deep ensemble with 5 ensemble members (Lakshminarayanan et al., 2017), and last-layer Laplace (Daxberger et al., 2021). The conventional baselines use standard data augmentation and were chosen because they support batch normalisation (Ioffe & Szegedy, 2015). We consider two variants of self-supervised BNNs: Self-Supervised BNNs pretrain using $\mathcal{D}^u$ only, while Self-Supervised BNNs* also use $\mathcal{D}^t$. Both variants use a non-linear projection head when pretraining and the data augmentations suggested by Chen et al. (2020a). We use a post-hoc Laplace approximation for the task-specific parameters (the last-layer parameters). We further consider ensembling self-supervised BNNs.

**Evaluation.** To evaluate the predictive performance of these BNNs, we report the negative log-likelihood (NLL). This is a proper scoring rule that simultaneously measures the calibration and accuracy of the different networks, and is thus an appropriate measure for *overall* predictive performance (Gneiting & Raftery, 2007). We also report the accuracy and expected calibration error (ECE) in Appendix Table D.1. We further assess out-of-distribution (OOD) generalisation from CIFAR10 to CIFAR10-C (Hendrycks & Dietterich, 2019). Moreover, we evaluate whether these BNNs can detect out-of-distribution inputs from SVHN (Netzer et al., 2011) when trained on CIFAR10. We report the area under the receiver operator curve (AUROC) metric using the predictive entropy. We want the OOD inputs from SVHN to have higher predictive entropies than the in-distribution inputs from CIFAR10.

**Results.** In Table 2, we report the NLL for each BNN when making predictions with different numbers of labelled examples. We see that self-supervised BNNs offer improved predictive performance over the baselines. In fact, on the full CIFAR-10 test set, a single self-supervised BNN* outperforms a deep ensemble, whilst being 5x cheaper when making predictions. We also show that self-supervised BNNs can also be ensembled to further improve their predictive performance. Incorporating the labelled training data during pretraining (SS BNN*) usually improves predictive performance. Self-supervised BNNs also offer strong performance out-of-distribution and consistently are able to perform out-of-distribution detection. Indeed, they are the only method with an AUROC exceeding 90% at all dataset sizes. In a further analysis in Appendix Table D.1, we also see that the improved NLL of self-supervised BNNs is in large part due to improvements in predictive accuracy, and that incorporating labelled data during pretraining also boosts accuracy. Overall, these results accord with our earlier findings about the improved prior predictives of self-supervised BNNs compared to standard BNNs, and highlight the substantial benefits of incorporating unlabelled data into the BNN pipeline.

Table 2: **BNN Predictive Performance**. We measure the performance of different BNNs for different numbers of labels. We consider in-distribution prediction, out-of-distribution (OOD) generalisation, and OOD detection. Shown is the mean and standard error across 3-5 seeds. The out-of-distribution generalisation results average over all corruptions with intensity level five from CIFAR10-C (Hendrycks & Dietterich, 2019). Recall that the SS BNN is performing the contrastive learning separately from and the SS BNN* jointly with the downstream task. We see that self-supervised BNNs offer improved predictive performance over conventional BNNs, especially in the low-data regime.

| Dataset | # labelled points | ↓ Negative Log Likelihood | | | | | | | |
| --- | --- | --- | --- | --- | --- | --- | --- | --- | --- |
| | | MAP | LL Laplace | SWAG | SS BNN | SS BNN* | Deep Ensemble | SS BNN Ensemble | SS BNN* Ensemble |
| CIFAR10 | 50 | $7.594_{\pm 1.092}$ | $2.259_{\pm 0.012}$ | $2.332_{\pm 0.005}$ | $1.047_{\pm 0.022}$ | $\mathbf{0.996}_{\pm 0.013}$ | $3.689_{\pm 0.174}$ | $0.980_{\pm 0.006}$ | $\mathbf{0.953}_{\pm 0.010}$ |
| | 500 | $2.504_{\pm 0.182}$ | $1.895_{\pm 0.020}$ | $2.072_{\pm 0.091}$ | $0.454_{\pm 0.004}$ | $\mathbf{0.441}_{\pm 0.004}$ | $1.805_{\pm 0.016}$ | $0.399_{\pm 0.001}$ | $\mathbf{0.384}_{\pm 0.002}$ |
| | 5000 | $1.570_{\pm 0.021}$ | $1.327_{\pm 0.042}$ | $1.028_{\pm 0.023}$ | $\mathbf{0.361}_{\pm 0.003}$ | $0.369_{\pm 0.013}$ | $0.846_{\pm 0.012}$ | $0.309_{\pm 0.001}$ | $\mathbf{0.292}_{\pm 0.002}$ |
| | 50000 | $0.613_{\pm 0.044}$ | $0.424_{\pm 0.013}$ | $0.312_{\pm 0.008}$ | $0.325_{\pm 0.005}$ | $\mathbf{0.256}_{\pm 0.002}$ | $0.272_{\pm 0.002}$ | $0.270_{\pm 0.001}$ | $\mathbf{0.204}_{\pm 0.001}$ |
| CIFAR100 | 50 | $11.86_{\pm 0.34}$ | $4.585_{\pm 0.006}$ | $6.840_{\pm 0.539}$ | $4.505_{\pm 0.002}$ | $\mathbf{4.496}_{\pm 0.002}$ | $10.38_{\pm 0.110}$ | $\mathbf{4.450}_{\pm 4e-4}$ | $4.492_{\pm 4e-4}$ |
| | 500 | $5.536_{\pm 0.060}$ | $4.359_{\pm 0.019}$ | $5.282_{\pm 0.285}$ | $2.640_{\pm 0.006}$ | $\mathbf{2.614}_{\pm 0.010}$ | $4.867_{\pm 0.007}$ | $2.533_{\pm 0.002}$ | $\mathbf{2.510}_{\pm 0.002}$ |
| | 5000 | $4.319_{\pm 0.163}$ | $3.362_{\pm 0.032}$ | $3.518_{\pm 0.169}$ | $\mathbf{1.689}_{\pm 0.003}$ | $1.910_{\pm 0.006}$ | $3.052_{\pm 0.009}$ | $\mathbf{1.524}_{\pm 0.001}$ | $1.644_{\pm 0.001}$ |
| | 50000 | $1.834_{\pm 0.064}$ | $1.469_{\pm 0.30}$ | $1.250_{\pm 0.010}$ | $1.435_{\pm 0.004}$ | $\mathbf{1.139}_{\pm 0.004}$ | $1.088_{\pm 0.012}$ | $1.212_{\pm 0.002}$ | $\mathbf{0.929}_{\pm 0.001}$ |
| CIFAR10 to CIFAR10-C (**OOD Generalisation**) | 50 | $7.140_{\pm 0.859}$ | $2.275_{\pm 0.006}$ | $2.353_{\pm 0.007}$ | $1.723_{\pm 0.004}$ | $\mathbf{1.697}_{\pm 0.010}$ | $3.970_{\pm 0.188}$ | $1.638_{\pm 0.006}$ | $\mathbf{1.603}_{\pm 0.004}$ |
| | 500 | $2.838_{\pm 0.175}$ | $2.045_{\pm 0.011}$ | $2.355_{\pm 0.083}$ | $1.272_{\pm 0.014}$ | $\mathbf{1.260}_{\pm 0.010}$ | $2.101_{\pm 0.021}$ | $1.164_{\pm 0.004}$ | $\mathbf{1.113}_{\pm 0.005}$ |
| | 5000 | $2.423_{\pm 0.267}$ | $1.644_{\pm 0.046}$ | $1.705_{\pm 0.084}$ | $\mathbf{1.235}_{\pm 0.007}$ | $1.237_{\pm 0.044}$ | $1.382_{\pm 0.023}$ | $1.103_{\pm 0.006}$ | $\mathbf{1.096}_{\pm 0.001}$ |
| | 50000 | $1.944_{\pm 0.223}$ | $\mathbf{1.244}_{\pm 0.067}$ | $\mathbf{1.215}_{\pm 0.051}$ | $1.287_{\pm 0.013}$ | $1.225_{\pm 0.014}$ | $\mathbf{0.984}_{\pm 0.026}$ | $1.126_{\pm 0.007}$ | $1.048_{\pm 0.004}$ |
| | | ↑ AUROC (%) | | | | | | | |
| CIFAR10 vs SVHN (**OOD Detection**) | 50 | $54.4_{\pm 4.53}$ | $48.4_{\pm 3.04}$ | $52.3_{\pm 1.37}$ | $87.1_{\pm 1.26}$ | $\mathbf{92.4}_{\pm 1.01}$ | $53.6_{\pm 2.42}$ | $\mathbf{91.3}_{\pm 0.25}$ | $90.9_{\pm 0.16}$ |
| | 500 | $61.2_{\pm 0.94}$ | $61.1_{\pm 1.19}$ | $51.1_{\pm 1.89}$ | $\mathbf{94.9}_{\pm 0.12}$ | $94.2_{\pm 0.45}$ | $62.1_{\pm 2.35}$ | $\mathbf{96.2}_{\pm 0.05}$ | $95.9_{\pm 0.07}$ |
| | 5000 | $83.3_{\pm 2.87}$ | $84.6_{\pm 0.63}$ | $59.6_{\pm 0.99}$ | $\mathbf{96.1}_{\pm 0.07}$ | $94.6_{\pm 1.00}$ | $92.9_{\pm 0.19}$ | $\mathbf{97.0}_{\pm 0.01}$ | $96.9_{\pm 0.12}$ |
| | 50000 | $93.8_{\pm 1.13}$ | $92.6_{\pm 2.01}$ | $76.4_{\pm 0.55}$ | $\mathbf{95.6}_{\pm 0.15}$ | $95.5_{\pm 0.16}$ | $96.8_{\pm 0.38}$ | $97.0_{\pm 0.05}$ | $\mathbf{97.7}_{\pm 0.06}$ |

Moreover, we perform an ablation of our variational distribution $q(W_j^c)$ in Appendix Table C.2, where we see that our data-dependent mean is indeed needed for good performance.

Note that our goal in this experiment is mainly to compare our self-supervised BNNs against other BNN methods on equal grounds, not necessarily to reach state-of-the-art performance on the used benchmark datasets. Indeed, reaching higher performances usually requires computationally expensive hyperparameter tuning (which we have not systematically performed) as well as using many engineering tricks, such as data augmentation and batch normalization. These tricks generally affect the likelihood in complicated ways and are thus often omitted from Bayesian neural networks (see, e.g., the discussions in Nabarro et al. (2021) and Krishnan et al. (2022)). This is why our results are empirically on par with many recent papers in Bayesian deep learning (e.g., Immer et al., 2021b; Ober & Aitchison, 2021; Izmailov et al., 2021b). However, it should be noted that some recent attempts have been made to reconcile BNNs with common practical deep learning tricks to reach high performance (c.f., Rudner et al., 2023). Adding these orthogonal ideas to our proposed framework would be a promising avenue for improving its performance to reach state-of-the-art levels.

## 5.2 Active Learning

We now highlight the benefit of incorporating unlabelled data in an active learning problem. We consider low-budget active learning, which simulates a scenario where labelling examples is extremely expensive. We use the CIFAR10 training set as the unlabelled pool set from which to label points. We assume an initial train set of 50 labelled points, randomly selected, and a validation set of the same size. We acquire 10 labels per acquisition round up to 500 labels and evaluate using the full test set. We compare self-supervised BNNs to a deep ensemble, the strongest BNN baseline. We use BALD (Houlsby et al., 2011) as the acquisition function for the deep ensemble and self-supervised BNN, which provide epistemic uncertainty estimates. We further compare to SimCLR using predictive entropy for acquisition because SimCLR does not model epistemic uncertainty.

In Fig. 4, we see that the methods that leverage unlabelled data perform the best. In particular, the self-supervised BNN with BALD acquisition achieves the highest accuracy across most numbers of labels, and substantially outperforms the deep ensemble. This confirms the benefit of incorporating unlabelled data in

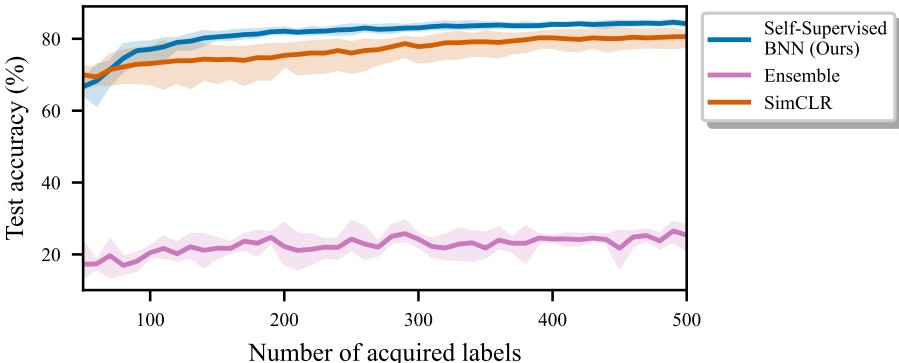

Figure 4: **Low-Budget Active Learning** on CIFAR10. We compare (i) a self-supervised BNN, (ii) SimCLR, and (iii) a deep ensemble. For the self-supervised BNN and the ensemble, we acquire points with BALD. We use predictive entropy for SimCLR, which does not provide epistemic uncertainty estimates. Mean and std. shown (3 seeds). The methods that incorporate unlabelled data perform best by far, with our method slightly outperforming SimCLR.

active learning settings, which by definition are semi-supervised and include unlabelled data. Moreover, our approach slightly outperforms SimCLR, suggesting that our Bayesian treatment of contrastive learning yields better uncertainties than conventional non-Bayesian contrastive learning. This is also confirmed in Appendix Fig. C.1, where we see that our approaches yield consistently lower calibration errors than SimCLR.

## 6 Related Work

**Improving BNN Priors.** We demonstrated that BNNs have poor prior predictive distributions (§4), a concern shared by others (e.g., Wenzel et al., 2020; Noci et al., 2021; Izmailov et al., 2021a). The most common approaches to remedy this are through designing better priors, typically over network parameters (Louizos et al., 2017; Nalisnick, 2018; Atanov et al., 2019; Fortuin et al., 2021b) or predictive functions directly (Sun et al., 2019; Tran et al., 2020; Matsubara et al., 2021; D'Angelo & Fortuin, 2021, see Fortuin (2022) for an overview). In contrast, our approach incorporates vast stores of unlabelled data into the prior distribution through variational model learning. Similarly, other work also *learns* priors, but typically using labelled data e.g., by using meta-learning (Garnelo et al., 2018; Rothfuss et al., 2021) or type-II maximum likelihood (Wilson et al., 2015; Immer et al., 2021a; Dhahri et al., 2024), or by using transfer learning in an *ad hoc* way (Shwartz-Ziv et al., 2022). Notably, function-space variational inference methods (Sun et al., 2019; Rudner et al., 2023) often also use unlabelled data, which has been shown to potentially improve the out-of-distribution performance of these models (Lin et al., 2023). However, in this case, the unlabelled data is only used for evaluating the KL divergence in function space, a practice which has theoretically been shown to be insufficient Burt et al. (2020). Conversely, our work uses semantic information from the unlabelled data to actually *inform* the function-space prior. Another related line of work is concerned with learning invariances from data in Bayesian models using the marginal likelihood (van der Wilk et al., 2018; Immer et al., 2022). This case is essentially the opposite of our setting, as there, the labels are known but the augmentations are learned, while in our case, the augmentations constitute our prior knowledge, but we do not know the data labels.

**A Perspective on Contrastive Learning.** We offer a Bayesian interpretation and understanding of contrastive learning (§3). Under our framework, pretraining is understood as model learning—a technique for finding probabilistic models with prior predictive distributions that capture our semantic beliefs. There has been much other work on understanding contrastive learning (e.g., Wang & Isola, 2020; Wang & Liu, 2021). Some appeal to the InfoMax principle (Becker & Hinton, 1992). Zimmermann et al. (2022) argue that contrastive learning inverts the data-generating process, while Aitchison (2021) cast InfoNCE as the objective

of a self-supervised variational auto-encoder. Ganev & Aitchison (2021) formulate several semi-supervised learning objectives as lower bounds of log-likelihoods in a probabilistic model of data curation.

**Semi-Supervised Deep Generative Models.** Deep generative models (DGMs) are a fundamentally different approach for label-efficient learning (Kingma & Welling, 2013; Kingma et al., 2014; Joy et al., 2020). A semi-supervised DGM models the full distribution $p(x, y)$ with generative modelling, and so incorporates unlabelled data by learning to generate it. Unlike BNNs, we can condition the parameters of a DGM on unlabelled data. In contrast, our approach does not model the data distribution—the unlabelled data is used to construct pseudo-labelled tasks that encode our prior beliefs. Self-supervised BNNs are discriminative models, which tend to be more scalable and perform better for discriminative tasks compared to full generative modelling (Ng & Jordan, 2001; Bouchard & Triggs, 2004). Finally, Sansone & Manhaeve (2022) try to unify self-supervised learning and generative modelling under one framework.

## 7 Conclusion

We introduced *Self-Supervised Bayesian Neural Networks*, which allow semantic information from unlabelled data to be incorporated into BNN priors. Using a novel evaluation scheme, we showed that self-supervised BNNs learn functional priors that better reflect the semantics of the data than conventional BNNs. In turn, they offer improved predictive performance over conventional BNNs, especially in low-data regimes. Going forward, we believe that effectively leveraging unlabelled data will be critical to the success of BNNs in many, if not most, potential applications. We hope our work encourages further development in this crucial area.

### Acknowledgments

MS was supported by the EPSRC Centre for Doctoral Training in Autonomous Intelligent Machines and Systems (EP/S024050/1), and thanks Rob Burbea for inspiration and support. VF was supported by a Postdoc Mobility Fellowship from the Swiss National Science Foundation, a Research Fellowship from St John's College Cambridge, and a Branco Weiss Fellowship.

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

# A   Self-Supervised BNNs: Further Considerations

We introduced *Self-Supervised BNNs* (§3), which benefit from unlabelled data for improved predictive performance within the probabilistic modelling framework. To summarise, our conceptual framework uses data augmentation to create a set of contrastive datasets $\{\mathcal{D}_j^c\}_{j=1}^L$. In our probabilistic model, conditioning on this data is equivalent to incorporating unlabelled data into the task prior predictive. We now discuss further considerations and provide further details.

**Theoretical Considerations.**   In §3.1, we treated the number of contrastive task datasets, $L$, as a fixed hyper-parameter. However, one could generate an potentially infinite number of datasets, in which case, the posterior $p(\theta^s|\{\mathcal{D}_j^c\}_{j=1}^L, \mathcal{D}^t) \propto p(\theta^s) \cdot p(\mathcal{D}^t|\theta^s) \cdot \prod_{j=1}^L p(\mathcal{D}_j^c|\theta^s)$ will collapse to a delta function, and will be dominated by the contrastive tasks. This justified learning a point estimate for $\theta^s$, but if one wanted to avoid this behaviour, one could re-define the posterior:

$$\tilde{p}(\theta^s|\{\mathcal{D}_j^c\}_{j=1}^L, \mathcal{D}^t) \propto p(\theta^s) \cdot p(\mathcal{D}^t|\theta^s) \cdot \prod_{j=1}^L p(\mathcal{D}_j^c|\theta^s)^{\gamma/L}. \tag{A.1}$$

The log-posterior would equal, up to a constant:

$$\log \tilde{p}(\theta^s|\{\mathcal{D}_j^c\}_{j=1}^L, \mathcal{D}^t) = \log p(\theta^s) + \log p(\mathcal{D}^t|\theta^s) + \frac{\gamma}{L} \sum_{j=1}^L \log p(\mathcal{D}_j^c|\theta^s). \tag{A.2}$$

Here, the total evidence contributed by the contrastive datasets is independent of $L$, and instead controlled by the hyper-parameter $\gamma$. This is equivalent to posterior tempering. The final term of the above equation could also be re-defined as an *average* log-likelihood when sampling different $\mathcal{D}^c$, i.e., we could use:

$$\log \tilde{p}(\theta^s|\mathcal{D}^u, \mathcal{D}^t) = \log p(\theta^s) + \log p(\mathcal{D}^t|\theta^s) + \gamma \mathbb{E}_{\mathcal{D}^c}[p(\mathcal{D}^c|\theta^s)]. \tag{A.3}$$

In this case, we look for a distribution over $\theta^s$ where $\mathcal{D}^c$ has a high likelihood on average. Our practical algorithm samples a different $\mathcal{D}^c$ per gradient step, and instead weights the $\mathcal{D}^t$ term with $\alpha$, which is similar to the above approach if we set $\alpha = 1/\gamma$ and modify the prior term as needed. Re-defining the framework in this way allows it to support potentially infinite numbers of generated contrastive datasets. We further note that this framework could naturally be extended to multi-task scenarios.

**Practical Considerations.**   In practice, we share parameters $\theta^s$ across all tasks (i.e., across the generated contrastive tasks and the actual downstream task), and learn a $\theta^t$ separately for each task. We focus on image classification problems and let $\theta^s$ be the parameters of a *base encoder*, which produces a representation $z$. $\theta^t$ are the parameters of a linear readout layer that makes predictions from $z$. We discussed learning a point-estimate for $\theta^s$ by optimising an ELBO derived from the unlabelled data only:

$$\tilde{\mathcal{L}}_j^c(\theta^s) = \mathbb{E}_{q(\theta_j^c)}[\log p(\mathcal{D}_j^c|\theta^s, \theta_j^c)] - D_{\mathrm{KL}}(q(\theta_j^c)||p(\theta_j^c)) \leq \log p(\mathcal{D}_j^c|\theta^s). \tag{A.4}$$

We (optionally) further include an ELBO derived using task-specific data:

$$\tilde{\mathcal{L}}^t(\theta^s) = \mathbb{E}_{q(\theta^t)}[\log p(\mathcal{D}^t|\theta^t, \theta^s)] - D_{\mathrm{KL}}[q(\theta^t)||p(\theta^t)] \leq \log p(\mathcal{D}^t|\theta^s). \tag{A.5}$$

Our final objective is:

$$\mathcal{L}(\theta^s) = \log p(\theta^s) + \alpha \tilde{\mathcal{L}}^t(\theta^s) + \mathbb{E}_{\mathcal{D}^c}[\tilde{\mathcal{L}}_j^c], \tag{A.6}$$

where $\alpha = 0$ would learn the base-encoder parameters only using the unlabelled data. $\alpha$ controls the weighting between the generated contrastive task data and the downstream data. We see the above objective is closely related to a (lower bound of) Eq. (A.3). We further modify this objective function, following best practice, either in the contrastive learning or Bayesian deep learning communities, and improve performance by:

1. We add a non-linear projection head to the base encoder architecture *only for the contrastive task datasets.* As such, we use $z = g_\phi(f_{\theta^s}(x))/||g_\phi(f_{\theta^s}(x))||$ for $\mathcal{D}^c$. $g_\phi(\cdot)$ is the projection head, and the representation is normalised. For the downstream tasks, we use $z = f_{\theta^s}(x)$, i.e., we "throw-away" the projection head. This is best practice within the contrastive learning community (Chen et al., 2020a;b).

2. We *temper* the KL divergence term, using the mean-per-parameter KL divergence (denoted as $\bar{D}_{\mathrm{KL}}(\cdot||\cdot)$). Tempering is necessary for several Bayesian deep learning algorithms to perform well (Wenzel et al., 2020; Krishnan et al., 2022).

3. We generate a new contrastive task dataset per gradient step, update on that dataset, and then discard it. This follows standard contrastive learning algorithms (Chen et al., 2020a).

4. We rescale the likelihood terms in the ELBOs $\tilde{\mathcal{L}}^t(\theta^s)$ and $\tilde{\mathcal{L}}_j^c(\theta^s)$ to be average per-datapoint log-likelihoods, e.g., we use $\frac{1}{|\mathcal{D}_j^c|} \log p(\mathcal{D}_j^c|\theta_j^t, \theta^s)$.

5. Instead of having an explicit prior distribution over $\theta^s$, we use standard weight-decay for training, i.e., we specify a penalty on the norm of the weights of the encoder *per gradient step.*

Together, these changes yield the objective function used in Algorithm 1. Finally, we note that different practical algorithms ensue depending on the choice of $\theta^t$, the choice of $\theta^s$, and the techniques used to perform approximate inference. We employ variational inference and learn a point estimate for $\theta^s$, but there are other choices possible.

# B  Experiment Details

We now provide further experiment details and additional results. The vast majority of experiments were run on an internal compute cluster using Nvidia Tesla V100 or A100 GPUs. The maximum runtime for an experiment was less than 16 hours.

## B.1  Semi-Supervised Learning (§5)

### B.1.1  Datasets

We consider the CIFAR10 and CIFAR100 datasets (Krizhevsky et al., 2009). The entire training set is the unsupervised set, and we suppose that we have access to different numbers of labels. For the evaluation protocols, we reserve a validation set of 1000 data points from the test set and evaluate using the remaining 9000 labels.

To assess out-of-distribution generalisation, we further evaluation on the CIFAR-10-C dataset (Hendrycks & Dietterich, 2018). We compute the average performance across all corruptions with intensity level five.

### B.1.2  Self-Supervised BNNs

**Base Architecture.**  We use a ResNet-18 architecture, modified for the size of CIFAR10 images, following Chen et al. (2020a). The representations produced by this architecture have dimensionality 512. Further, for the non-linear projection head, we use a 2 layer multi-layer perceptron (MLP) with output dimensionality 128.

**Contrastive Augmentations.**  We follow Chen et al. (2020a) and compose a random resized crop, a random horizontal flip, random colour jitter, and random grayscale for the augmentation. These augmentations make up the contrastive augmentation set $\mathcal{A}$. We finally normalise the images to have mean 0 and standard deviation 1 per channel, as is standard.

**Hyperparameters.**  We use a $\mathcal{N}(0, \frac{1}{\tau_p^2})$ prior over the linear parameters $\theta^t$, and tune $\tau_p$ for each dataset. As such, $\tau$ can be understood as the prior temperature. We use $\tau_p = 0.65$ for CIFAR10 and $\tau_p = 0.6$ for CIFAR100. We use weight decay $1e-6$ for the base encoder and projection head parameters.

**Variational Distribution.** We parameterize the temperature and noise scale using the log of their values. That is, we have $\sigma = \exp \tilde{\sigma}$.

**Optimisation Details.** We use the LARS optimiser (You et al., 2017), with batch size 1000 and momentum 0.9. We train for 1000 epochs, using a linear warmup cosine annealing learning rate schedule. The warmup starting learning rate for the base encoder parameters is $1e-3$ with a maximum learning rate of 0.6. For the variational parameters, the maximum learning rate is $1e-3$, which we found to be important for the stability of the algorithm.

**Laplace Evaluation Protocol.** We find a point estimate for $\theta^t$ found using the standard linear evaluation protocol (i.e., SGD training). We then apply a post-hoc Laplace approximation using the generalised Gauss-Newton approximation to the Hessian using the `laplace` library (Daxberger et al., 2021). For CIFAR10, we use a full covariance approximation and for CIFAR100 we use a Kroenker-factorised approximation for the Hessian of the last layer weights and biases. We tune the prior precision by maximising the likelihood of a validation set. For predictions, we use the (extended) probit approximation. These choices follow recommendations from Daxberger et al. (2021).

### B.1.3  Self-Supervised BNNs*

Self-Supervised BNNs* additionally leverage labelled data when training the base encoder by including an additional ELBO $\tilde{\mathcal{L}}^t$ that depends on $\mathcal{D}^t$.

We use mean-field variational inference over $\theta^t$ with a Gaussian approximate posterior and Flipout (Wen et al., 2018). We use the implementation from Krishnan et al. (2022), and temper by setting $\beta = 1/|\theta^t|$, meaning we use the average per-parameter KL divergence. $\alpha$ is a hyperparameter that controls the relative weighting between the generated contrastive task datasets and the observed label data, and is tuned. For CIFAR10, We use $\alpha = 5 \cdot 10^{-5}$ when we have fewer than 100 labels, and $\alpha = 5 \cdot 10^{-3}$ otherwise. For CIFAR100, We use $\alpha = 5 \cdot 10^{-5}$ when we have fewer than 1000 labels, and $\alpha = 5 \cdot 10^{-3}$ otherwise. For $p(\theta^t)$, we use a $\mathcal{N}(0,1)$ prior. For downstream evaluation, we use the Laplace evaluation protocol.

All other details follow Self-Supervised BNNs.

### B.1.4  BNN Baselines

All baselines use the same ResNet-18 architecture, which was modified for the image size used in the CIFAR image datasets. The baselines we considered were chosen because they are all compatible with batch normalisation, which is included in the base architecture. We provide further details about the baselines below.

**MAP.** For the maximum-a-posterior network, we use the Adam optimiser with learning rate $10^{-3}$, default weight decay, and batch size 1000. We train for a minimum of 25 epochs and a maximum of 300 epochs, terminating training early if the validation loss increases for 3 epochs in a row.

**Last-Layer Laplace.** For the Last-Layer Laplace baseline, we perform a post-hoc Laplace approximation to a MAP network trained using the protocol above. We use the same settings as for the self-supervised BNN's Laplace evaluation.

**Deep Ensemble.** For the deep ensemble baseline, we train 5 MAP networks starting from different initialisations using the above protocol, and aggregate their predictions.

**SWAG.** For the SWAG baseline, we first a MAP network using the above protocol. We then run SGD from this solution for 10 epochs, taking 4 snapshots per epoch, and using $K = 20$ as the rank of the covariance matrix. We choose the SWAG learning rate per run using the validation set, and consider $10^{-2}, 10^{-3}$, and $10^{-4}$.

### B.2  Active Learning (§5)

We simulate a low-budget active learning setting. For each method, we use their default implementation details as outlined in this Appendix. With regards to the active learning setup, we assume that we have access to a small validation set of 50 labelled examples and are provided 50 labelled training examples. We acquire 10 examples per acquisition round up to a maximum of 500 labelled examples, which corresponds to 1% of the labels in the training set. We evaluate using the full test set. The deep ensemble and self-supervised BNNs provide epistemic uncertainty estimates, so we perform active learning by selecting the points with the highest BALD metric (Houlsby et al., 2011). For SimCLR, we acquire points using the highest predictive entropy, a commonly used baseline (Gal et al., 2017). For this experiment, we use only the CIFAR10 dataset. SimCLR and the self-supervised BNNs here are pretrained on 500 epochs, not 1000 epochs as default.

### B.3  Prior Predictive Checks (§4)

**BNN Prior Predictive.**   We use the ResNet-20-FRN architecture, which is the architecture used by Izmailov et al. (2021b). Note that this architecture does not include batch normalisation, which means the prior over parameters straightforwardly corresponds to a prior predictive distribution. We use a $\mathcal{N}(0, \frac{1}{5})$ prior over all network weights, again following Izmailov et al. (2021b), and sample from the prior predictive using 8192 Monte Carlo samples.

**Self-Supervised Prior Predictives.**   We primarily follow the details outlined earlier, except we use the same ResNet-20-FRN architecture as used for the BNNs and batch size of 500 rather than 1000. To sample from the prior predictive, we use Eq. (2) and have $y \sim \text{softmax}(W f_{\theta^s}(x))$, where we normalise the representations produced by the base encoder to have zero mean and unit variance, and we have $W \sim \mathcal{N}(0, 20)$, with the prior precision chosen by hand. We neglect the biases because they introduce additional variance. The prior evaluation scores are not sensitive to the prior variance choice, and are evaluated by sampling images from the validation set, which was not seen during training. We used 4096 Monte Carlo samples from the prior.

## C  Ablation Studies

**Effect of Batch Size.**  We study the effect of the pre-training batch size on the performance of our self-supervised BNNs. We run one seed for 100 epochs across three different batch sizes on CIFAR10. We see in Table C.1, the performance on CIFAR10 is robust to reducing the batch size. We hypothesise this is due to the noise injected during pre-training.

Table C.1: Effect of pretraining batch size on self-supervised BNN.

| Batch Size | CIFAR10 Accuracy (%) |
|---|---|
| 100 | 0.81 |
| 500 | 0.81 |
| 1000 | 0.80 |

**Effect of Variational Distribution.**  We run an ablation study changing the variational distribution mean on CIFAR10. We evaluated using one seed, training for 100 epochs only. We consider setting the mean of the variational distribution for image $i$, $\omega_i$, to be: $0.5(\tilde{z}_i^A + \tilde{z}_i^B)$, $\tilde{z}_i^A$, and $\mathbf{0}$. We see that a suitable mean is required for good performance.

Table C.2: Effect of the pretraining variational distribution on self-supervised BNN performance on CIFAR10. We see that some variant of our data-dependent mean is needed for good performance.

| Variational Dist. Mean | CIFAR10 Accuracy (%) |
|---|---|
| $\mathbf{0}$ | 0.19 |
| $\tilde{z}_i^A$ | 0.79 |
| $0.5(\tilde{z}_i^A + \tilde{z}_i^B)$ | 0.80 |

**Effect of Pretraining and Inference.**  To better understand the effect of the pretraining objective and the approximate inference scheme used, we performed an ablation study on CIFAR10. We considered both variants of our variational pretraining, and additionally deterministic pretraining that uses the NT-XENT loss. We also consider either using Laplace approximate inference or MAP estimation for the task parameters. Using the NT-XENT loss and MAP inference corresponds to SimCLR (Chen et al., 2020a), as widely used in the self-supervised learning community. For NT-XENT, we use $\tau = 0.45$ for CIFAR10 and $\tau = 0.3$ for CIFAR100. For these experiments, we only train for 500 epochs.

In Fig. C.1, we see that incorporating the labelled data during pretraining boosts accuracy, but surprisingly decreases calibration. Relative to SimCLR, all of the self-supervised BNNs offer improved calibration at all dataset sizes. All approaches have high accuracy at low data regimes, highlighting the benefit of leveraging unlabelled data. Both deterministic pretraining and variational pretraining behave similarly, but performing approximate inference over task parameters substantially improves calibration.

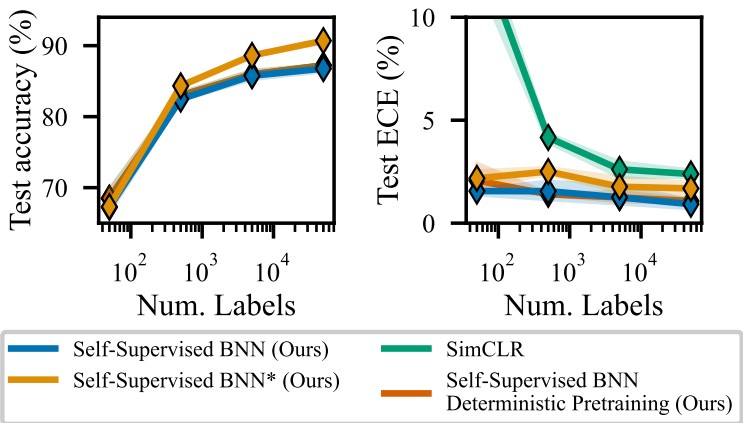

Figure C.1: **Effect of Pretraining and Inference** on CIFAR10. On the left plot, red, green, and blue lines overlap. On the right plot, blue and red lines overlap. Recall that the SS BNN is performing the contrastive learning separately from and the SS BNN* jointly with the downstream task. We see that our SS BNN* slightly outperforms the other approaches in terms of accuracy and that all of our approaches yield better-calibrated uncertainties than SimCLR.

## D  Additional Results

We additionally report the in-distribution accuracy and expected calibration error (ECE) of different BNNs when observing different numbers of labels. These metrics, unlike the log-likelihood, are interpretable. But note that for a useful classifier, we need to have *both* accurate *and* well-calibrated predictions.[2]

In Table D.1, we see that self-supervised BNNs substantially outperform conventional BNNs in terms of in-distribution accuracy. The gains are particularly large at smaller dataset sizes, precisely where improved priors are expected to make the biggest difference. Moreover, in terms of calibration, they consistently offer well-calibrated uncertainty estimates. Even though LL Laplace offers well-calibrated uncertainty estimates at a low numbers of labels, the predictions are much less accurate than self-supervised BNNs. We also see that incorporating labelled data during pretraining or ensembling self-supervised BNNs boosts accuracy, but surprisingly can harm calibration. Curiously, we find that the calibration of ensemble methods also sometimes worsens as we condition on more data.

---

[2]There are perfectly calibrated but useless classifiers, e.g., if 70% of examples are class A and 30% are class B, predicting $p(\text{class A}) = 0.7$ on every input achieves perfect ECE but does not discriminate between examples at all.

Table D.1: **Bayesian Neural Network Predictive Performance**. Here, relative to the main results table, we report the accuracy and expected calibration error of different methods separately. Recall that the SS BNN is performing the contrastive learning separately from and the SS BNN* jointly with the downstream task.

| Dataset | # labelled points | ↑ Accuracy (%) | | | | | | | |
|---|---|---|---|---|---|---|---|---|---|
| | | MAP | LL Laplace | SWAG | SS BNN | SS BNN* | Deep Ensemble | SS BNN Ensemble | SS BNN* Ensemble |
| CIFAR10 | 50 | $14.6_{\pm0.7}$ | $14.9_{\pm0.8}$ | $14.8_{\pm0.2}$ | $66.3_{\pm0.8}$ | $\mathbf{68.3}_{\pm0.2}$ | $19.0_{\pm0.3}$ | $68.9_{\pm0.1}$ | $\mathbf{69.5}_{\pm0.2}$ |
| | 500 | $31.2_{\pm0.4}$ | $32.4_{\pm0.9}$ | $31.5_{\pm7.4}$ | $84.8_{\pm0.2}$ | $\mathbf{86.2}_{\pm0.1}$ | $39.2_{\pm0.5}$ | $87.0_{\pm0.1}$ | $\mathbf{87.6}_{\pm0.1}$ |
| | 5000 | $56.5_{\pm3.1}$ | $53.4_{\pm2.4}$ | $66.4_{\pm1.2}$ | $87.7_{\pm0.1}$ | $\mathbf{88.6}_{\pm0.2}$ | $72.1_{\pm0.3}$ | $89.6_{\pm0.2}$ | $\mathbf{90.9}_{\pm0.03}$ |
| | 50000 | $83.4_{\pm1.9}$ | $85.7_{\pm0.3}$ | $90.5_{\pm0.6}$ | $88.6_{\pm0.1}$ | $\mathbf{91.7}_{\pm0.1}$ | $91.8_{\pm0.3}$ | $90.7_{\pm0.2}$ | $\mathbf{93.2}_{\pm0.07}$ |
| CIFAR100 | 50 | $3.6_{\pm0.2}$ | $3.6_{\pm0.3}$ | $3.1_{\pm0.6}$ | $\mathbf{14.5}_{\pm0.1}$ | $14.7_{\pm0.3}$ | $3.8_{\pm0.06}$ | $\mathbf{16.2}_{\pm0.02}$ | $16.5_{\pm0.1}$ |
| | 500 | $7.0_{\pm0.2}$ | $6.1_{\pm0.2}$ | $7.7_{\pm0.3}$ | $\mathbf{38.5}_{\pm0.2}$ | $39.0_{\pm0.3}$ | $9.2_{\pm0.1}$ | $40.8_{\pm0.1}$ | $\mathbf{41.3}_{\pm0.01}$ |
| | 5000 | $22.7_{\pm0.7}$ | $21.2_{\pm0.6}$ | $25.6_{\pm0.9}$ | $54.5_{\pm0.1}$ | $\mathbf{56.0}_{\pm0.3}$ | $28.2_{\pm0.1}$ | $58.4_{\pm0.1}$ | $\mathbf{62.5}_{\pm0.1}$ |
| | 50000 | $60.5_{\pm2.3}$ | $59.6_{\pm1.1}$ | $67.6_{\pm0.5}$ | $59.9_{\pm0.1}$ | $\mathbf{69.2}_{\pm0.1}$ | $70.7_{\pm0.5}$ | $66.4_{\pm0.1}$ | $\mathbf{74.9}_{\pm0.1}$ |
| CIFAR10 to CIFAR10-C (**OOD Generalisation**) | 50 | $13.6_{\pm0.7}$ | $13.8_{\pm0.7}$ | $14.0_{\pm1.6}$ | $\mathbf{45.5}_{\pm0.03}$ | $45.5_{\pm0.5}$ | $16.9_{\pm0.4}$ | $47.7_{\pm0.1}$ | $\mathbf{48.0}_{\pm0.1}$ |
| | 500 | $26.7_{\pm0.4}$ | $27.2_{\pm0.8}$ | $26.2_{\pm5.2}$ | $57.8_{\pm0.4}$ | $\mathbf{59.0}_{\pm0.2}$ | $32.0_{\pm0.5}$ | $60.3_{\pm0.1}$ | $\mathbf{61.8}_{\pm0.1}$ |
| | 5000 | $42.1_{\pm0.9}$ | $43.2_{\pm1.1}$ | $50.5_{\pm1.6}$ | $59.6_{\pm0.1}$ | $\mathbf{60.3}_{\pm1.1}$ | $55.1_{\pm0.3}$ | $62.9_{\pm0.2}$ | $\mathbf{64.0}_{\pm0.3}$ |
| | 50000 | $59.2_{\pm3.7}$ | $62.1_{\pm1.6}$ | $\mathbf{69.0}_{\pm0.4}$ | $59.8_{\pm0.3}$ | $63.1_{\pm0.5}$ | $\mathbf{70.1}_{\pm0.4}$ | $63.6_{\pm0.2}$ | $66.8_{\pm0.2}$ |
| | | ↓ Expected Calibration Error (ECE; %) | | | | | | | |
| CIFAR10 | 50 | $65.9_{\pm3.2}$ | $2.7_{\pm0.7}$ | $6.2_{\pm2.0}$ | $\mathbf{1.7}_{\pm0.02}$ | $1.9_{\pm0.3}$ | $35.3_{\pm2.8}$ | $\mathbf{1.9}_{\pm0.1}$ | $2.0_{\pm0.2}$ |
| | 500 | $30.0_{\pm2.1}$ | $3.2_{\pm0.6}$ | $13.4_{\pm0.6}$ | $\mathbf{1.5}_{\pm0.2}$ | $2.6_{\pm0.1}$ | $10.6_{\pm0.6}$ | $2.3_{\pm0.1}$ | $\mathbf{2.0}_{\pm0.1}$ |
| | 5000 | $21.0_{\pm0.5}$ | $4.5_{\pm0.4}$ | $10.0_{\pm0.3}$ | $\mathbf{1.1}_{\pm0.08}$ | $2.1_{\pm0.1}$ | $3.8_{\pm0.6}$ | $\mathbf{2.3}_{\pm0.1}$ | $2.6_{\pm0.2}$ |
| | 50000 | $8.4_{\pm0.5}$ | $1.5_{\pm0.4}$ | $2.8_{\pm0.4}$ | $\mathbf{0.8}_{\pm0.02}$ | $1.2_{\pm0.1}$ | $3.7_{\pm0.3}$ | $2.8_{\pm0.1}$ | $2.1_{\pm0.1}$ |
| CIFAR100 | 50 | $61.6_{\pm1.1}$ | - | $17.5_{\pm6.0}$ | $\mathbf{11.6}_{\pm0.1}$ | $11.7_{\pm0.3}$ | $38.5_{\pm0.7}$ | $\mathbf{13.4}_{\pm0.02}$ | $13.6_{\pm0.1}$ |
| | 500 | $25.3_{\pm0.7}$ | $\mathbf{1.8}_{\pm1.1}$ | $22.0_{\pm4.5}$ | $2.1_{\pm0.1}$ | $2.7_{\pm0.3}$ | $14.9_{\pm0.2}$ | $3.3_{\pm0.04}$ | $\mathbf{2.9}_{\pm0.1}$ |
| | 5000 | $32.3_{\pm2.1}$ | $\mathbf{1.4}_{\pm0.06}$ | $16.4_{\pm6.4}$ | $1.5_{\pm0.2}$ | $7.8_{\pm0.3}$ | $4.4_{\pm0.5}$ | $\mathbf{3.2}_{\pm0.1}$ | $12.3_{\pm0.2}$ |
| | 50000 | $17.8_{\pm2.2}$ | $\mathbf{1.6}_{\pm0.2}$ | $8.9_{\pm0.9}$ | $1.7_{\pm0.1}$ | $2.5_{\pm0.1}$ | $\mathbf{4.7}_{\pm0.08}$ | $6.5_{\pm0.1}$ | $6.6_{\pm0.1}$ |
| CIFAR10 to CIFAR10-C (**OOD Generalisation**) | 50 | $67.6_{\pm3.0}$ | $\mathbf{4.0}_{\pm0.7}$ | $6.9_{\pm0.8}$ | $8.4_{\pm0.6}$ | $9.2_{\pm0.7}$ | $37.3_{\pm2.9}$ | $6.1_{\pm0.1}$ | $\mathbf{5.9}_{\pm0.1}$ |
| | 500 | $33.7_{\pm1.9}$ | $\mathbf{7.4}_{\pm0.8}$ | $18.0_{\pm8.0}$ | $8.1_{\pm0.7}$ | $8.9_{\pm0.6}$ | $15.5_{\pm0.3}$ | $\mathbf{7.2}_{\pm0.1}$ | $7.6_{\pm0.4}$ |
| | 5000 | $30.4_{\pm2.6}$ | $9.2_{\pm1.3}$ | $19.1_{\pm0.8}$ | $7.4_{\pm0.4}$ | $7.8_{\pm0.8}$ | $6.9_{\pm0.3}$ | $\mathbf{5.0}_{\pm0.1}$ | $6.5_{\pm0.2}$ |
| | 50000 | $24.1_{\pm3.0}$ | $11.1_{\pm0.6}$ | $13.6_{\pm1.3}$ | $\mathbf{9.1}_{\pm0.2}$ | $12.2_{\pm0.5}$ | $\mathbf{6.0}_{\pm0.6}$ | $6.1_{\pm0.1}$ | $6.8_{\pm0.01}$ |

