# OpenReview forum: "Incorporating Unlabelled Data into Bayesian Neural Networks"
_TMLR — Accepted by TMLR_

### Review · Reviewer_Eww4 · 2024-02-24

**Summary Of Contributions:**

The paper proposes using unlabelled data to pretrain a BNN prior for greater sample efficiency in downstream tasks.
They propose a pre-training scheme that uses a contrastive-style ELBO objective combined with data augmentation.

They look at image classification, out-of-distribution detection, and active learning.

**Audience:**

Yes

**Claims And Evidence:**

No

**Requested Changes:**

Based on the feedback above, my major concerns can be summarized as follows:

1) Retract claims on BNNs and unlabelled data w.r.t. FVI and discuss FVI with greater accuracy

2) Disentangle architecture and approximate inference in the experimental section

3) Rewrite paper to motivate instead by learning invariant feature spaces rather than learning weight prior distributions, or do experiments that learn weight priors for the pretrained network torso

4) Remove Section 4 and use Figure 3 and Table 1 as evidence for the behavior of Algorithm 1.

**Strengths And Weaknesses:**

**Strengths**

One of the motivations of Bayesian methods is that the priors should be a good initialization for a given ML task. It makes sense to learn priors that allow fast adaption to tasks in the low-data regime.
The proposed approach does appear to show strong sample-efficiency in the experiments, in large-ish scale experiment settings such as CIFAR 10. It is also good at out-of-distribution and active learning.

**Weaknesses**

**Function-space VI.**
Throughout the paper, the authors claim that BNN methods cannot use unlabeled data. In functional / function-space VI (Sun et al. 2019, which the authors cite), the function-space KL is enforced by a 'measurement set'. This measurement set is essentially samples from the data domain, so it could easily include unlabelled data. In fact, there was an AABI paper last year that used FVI to improve OOD performance of BNN using unlabelled OOD data [A]. The authors need to revise their claims and potentially compare them to FVI methods if they are relevant.

**Actual relevance to BNNs.**
By the end of the paper, I was questioning how relevant this paper was to actual *Bayesian* NNs and not just NNs more generally. Since it seems the shared 'torso' network is trained to obtain a point estimate of the shared parameters, it's not really an interesting prior distribution over the weights, and the initial motivation of the work feels a bit misleading. You could focus the paper on the value of this pre-trained torso network and just use BNNs for the evaluation.
At the very least, I would motivate the work through features /kernels rather than weight priors as done in the introduction.

**Prior design and inventing a metric that the method was designed to solve.**
It's not a good idea to invent a new metric to evaluate a model since there is a strong motivation to bias it towards the proposed model's strengths.
I don't agree that 'similar inputs should predict the same class' is a sound prior we want models to have, and Section 4 seems to suggest it's an inherently good idea when it is, in fact, a subjective design decision made by the authors.
I think a good BNN prior for classification should pick no specific class, and each input should have a close-to-uniform distribution over labels.
This is the motivation for several works using stationary Gaussian processes-like architectures for BNNs [B, C].
The authors could consider combining their method with the architecture of [B, C] to get the best of both worlds.
In summary, the authors have conflated a model having invariances with a model's prior, which are not exactly the same. A model can have invariances without it necessarily being evident in it's prior. For example, you could use sine and cosine features for angular data in a stationary Gaussian process to encode rotational invariance, but this doesn't change the stationary nature of the prior and the predictive distribution is unchanged.

I think a more principled metric would be the 'effective dimension' (c.f. [D]) of the kernel induced by the pre-trained feature space.
For $n$ data points, the effective dimension of the kernel (basically the rank of the data matrix, with the kernel being the outer product of the features) is $n$ if the features are unique and $1$ if the feature /data point is just repeated $n$ times, therefore minimizing the effective dimension for augmented inputs would be a more principled way of encoding this prior knowledge into the model and focuses on the fact that the authors are optimizing a deterministic feature space rather than a weight distribution.

**The neural linear model.**
The authors don't seem to know about the neural linear model / Bayesian last layer, which is a Bayesian linear regression on a neural network feature space. It's one of the oldest approaches to BNNs [E] and has lots of attractive properties, such as tractable marginal likelihoods in the regression settings [F] and also tractable ELBOs [G]. Throughout the paper, the authors refer to 'last-layer Laplace' and 'partial stochasticity' when the neural linear model is an older and more explicit way of describing this approach.

**Invariance learning with the marginal likelihood.** I'm not very familiar with the body of work, but I know that the marginal likelihood has been used to train invariances with Bayesian neural networks [H, I]. This body of work seems relevant enough to discuss as related work, and even relevant enough to evaluate against it since the pre-trained feature space is also essentially learning invariances.

**The experiments mix up approximate inference with architectures.**
The experimental evaluation, while thorough, was a bit confused. The central question that was tackled by the experiments was 'Does pre-training the feature space improve BNN uncertainty quantification?'. This is invariant to the choice of approximate inference for the BNN itself. Therefore, the experiments could rather use the pre-trained network as the torso for the MAP, SWAG and ensemble and see if the torso brings a net benefit across approximate inference methods. Comparing two very different architectures and approximate inference methods is not that meaningful.

**Clarity**
A few clarity issues:
* The ''# Labels' column in Table 2 is a bit confusing since I believe it's referring to the number of training data points, but it reads like the dimensionality of the classification problem is increasing
*  The definition of SS BNN* should be repeated a few times (e.g. the caption of Table 2) because the name is vague and it's easy to miss/forget while reading the main text.

[A] Function-Space Regularization for Deep Bayesian Classification, Lin et al.

[B] Periodic Activation Functions Induce Stationarity, Meronen et al

[C] Simple and Principled Uncertainty Estimation with Deterministic Deep Learning via Distance Awareness, Liu et al

[D] Bandit optimisation of functions in the Matern kernel RKHS, Janz et al

[E] Marginalized neural network mixtures for large-scale regression, Lázaro-Gredilla et al

[F] Benchmarking the Neural Linear Model for Regression, Ober et al

[G] Variational Bayesian Last Layers, Harrison et al

[H] Learning Invariances using the Marginal Likelihood, van der Wilk et al

[I] Invariance Learning in Deep Neural Networks with Differentiable Laplace Approximations, Immer et al

---

> ### Author Response · Authors · 2024-04-12
> **Response 1/2**
>
> We would like to thank you for your insightful comments and questions, which have truly helped us improve the paper. We will address your concerns in the following. Note that revisions in our updated manuscript are highlighted in green.
> We are encouraged to see that you find our proposed approach sensible and our empirical results convincing.
>
> > Throughout the paper, the authors claim that BNN methods cannot use unlabeled data. In functional / function-space VI (Sun et al. 2019, which the authors cite), the function-space KL is enforced by a 'measurement set'. This measurement set is essentially samples from the data domain, so it could easily include unlabelled data. In fact, there was an AABI paper last year that used FVI to improve OOD performance of BNN using unlabelled OOD data [A].
>
>
> Thank you for these pointers. Indeed, these works use unlabelled data to evaluate the function-space KL divergence, which can improve the model's ability to recognize OOD data, where the unlabeled data takes the role of representing OOD data in contrast to the ID labelled data that the model is trained on.
> In our work, we use unlabeled data to allow the model to learn about the semantics of the data space, by virtue of using contrastive tasks on aumentations of the unlabeled data. In this case, the unlabeled data is meant to be representative of our ID data and the contrastive tasks should share semantic structure with the actual labels. Our approach therefore is meant to improve ID performance of the models, while we would not foresee it to necessarily improve OOD detection.
> Note also that the use of a finite measurement set for evaluating the KL divergence in function-space VI has been seriously criticized on theoretical grounds [1].
> We have added a discussion of this to the manuscript.
>
> [1] https://arxiv.org/abs/2011.09421
>
>
> > Since it seems the shared 'torso' network is trained to obtain a point estimate of the shared parameters, it's not really an interesting prior distribution over the weights, and the initial motivation of the work feels a bit misleading. You could focus the paper on the value of this pre-trained torso network and just use BNNs for the evaluation. At the very least, I would motivate the work through features /kernels rather than weight priors as done in the introduction.
>
> We agree that the learned point estimate of $\theta^s$ should not be thought of as a weight space prior, but rather as a function-space prior. It (hopefully usefully) biases the space of functions that the BNN can represent in the end. We also like your suggestion of thinking of this in terms of a kernel, which makes the similarity to explicit function-space priors, such as Gaussian processes, clear. We have added a discussion of this to the manuscript.
>
>
> >  I don't agree that 'similar inputs should predict the same class' is a sound prior we want models to have, and Section 4 seems to suggest it's an inherently good idea when it is, in fact, a subjective design decision made by the authors.
>
> This is an interesting point. Indeed, it could be argued that there may be tasks where this is not a useful prior. In our work, we focus on tasks where contrastive learning has shown great performance, such as in computer vision. Approaches such as SimCLR have yielded great advances there, presumably because this kind of prior knowledge is really helpful there. Our main goal in this work was to show that the kind of prior knowledge exploited by SimCLR can also be incorporated in a Bayesian context. We have clarified this limitation in the manuscript.
>
>
> >  In summary, the authors have conflated a model having invariances with a model's prior, which are not exactly the same. A model can have invariances without it necessarily being evident in its prior. For example, you could use sine and cosine features for angular data in a stationary Gaussian process to encode rotational invariance, but this doesn't change the stationary nature of the prior and the predictive distribution is unchanged.
>
> Thank you for this point, it is a very insightful way of looking at it. Indeed, our approach takes invariances in the form of known image augmentations and incorporates these invariances into the functional prior of the BNN. We agree that in some cases, invariances can be directly encoded via feature engineering as you suggest. However, in the case of these complex semantic invariances of images, it seems more prudent to incorporate them into the prior using our framework, since it is hard to see how one would go about designing such features by hand. We have added this interesting discussion to our manuscript.
>
>
> > Throughout the paper, the authors refer to 'last-layer Laplace' and 'partial stochasticity' when the neural linear model is an older and more explicit way of describing this approach.
>
> Thanks for this pointer, we added references to the neural linear model to the manuscript.

---

> > ### Author Response · Authors · 2024-04-12
> > **Response 2/2**
> >
> > > I know that the marginal likelihood has been used to train invariances with Bayesian neural networks [H, I]. This body of work seems relevant enough to discuss as related work, and even relevant enough to evaluate against it since the pre-trained feature space is also essentially learning invariances.
> >
> > Thanks also for this interesting pointer. In these works, the authors do not know the invariances, but have access to the labels and use them to learn the invariances (in the form of data augmentations) from data. Our approach is essentially exactly the opposite: we know the data augmentations a priori, but do not have access to the data labels, and incorporate the invariances into the prior of the model to learn more efficiently with fewer labels. We have added a discussion of this line of work to the paper.
> >
> >
> > > The experiments mix up approximate inference with architectures. [...] Comparing two very different architectures and approximate inference methods is not that meaningful.
> >
> > We would like to clarify that the architecture for the evaluation on the supervised task is always the same in our experiments, e.g., for CIFAR-10, it is always a ResNet-18. Our main goal in the experiments was to provide baselines of BNN methods that people would actually use in practice (without pre-training) and show that our pretraining approach improves performance compared to them. So as you correctly point out, the message should be "pre-training the prior helps", not "pre-training is better than Laplace", which would indeed be an apples-to-oranges comparison. In our prior-learning framework, we use Laplace inference at prediction time, but we would indeed expect similar improvements when using other types of inference.
> >
> > > The ''# Labels' column in Table 2 is a bit confusing since I believe it's referring to the number of training data points, but it reads like the dimensionality of the classification problem is increasing
> >
> > Thanks, we have renamed that column to "# labelled points" to make it clearer.
> >
> > > The definition of SS BNN* should be repeated a few times (e.g. the caption of Table 2) because the name is vague and it's easy to miss/forget while reading the main text.
> >
> > You are right, we have repeated it now in different places.
> >
> >
> > We would like to thank you again for your comments and hope to have addressed all your concerns. Please let us know if you have any further questions. Moreover, in the spirit of TMLR, we are open to modifying our claims to reflect our presented evidence better, if you believe that that would be necessary.

---

### Review · Reviewer_R7PH · 2024-03-06

**Summary Of Contributions:**

The paper proposes a way to perform semi-supervised learning of Bayesian Neural Networks (BNNs).
The proposed approach mostly follows the contrastive learning technique from [1], i.e. the authors train an encoder on the labeled data via the constrastive loss, which then can be used for supervised learning on downstream tasks with labeled data. The main focus of the paper is on bringing the Bayesian perspective to this framework. Namely, the authors propose to use Bayesian linear layers as "heads" for both the contrastive learning and downstream tasks. The encoder model precisely follows [1], though.

For the empirical evaluation, the authors evaluate their approach on CIFAR-10 and CIFAR-100, from which they partially remove the labels. For the baselines, the authors choose several Bayesian methods that do not incorporate unlabelled data into their training procedure.

[1] Chen, Ting, Simon Kornblith, Mohammad Norouzi, and Geoffrey Hinton. "A simple framework for contrastive learning of visual representations." In International conference on machine learning, pp. 1597-1607. PMLR, 2020.

**Audience:**

No

**Broader Impact Concerns:**

The paper does not require including a broader impact statement.

**Claims And Evidence:**

No

**Requested Changes:**

The paper requires a major revision. See the list of major concerns for the requested changes.

**Strengths And Weaknesses:**

I have several major concerns regarding the contrastive learning procedure considered in the paper.
1. The contrastive loss is not specified in the paper. The only description of the loss function is as follows.
```
The contrastive task is to predict the corresponding source image index for each augmented image. That is, for the first pair of augmented images in a given contrastive task dataset, we want to predict class “1”, for the second pair, we want to predict class “2”, and so forth. ... To predict these labels, we use a linear layer applied to an encoder that produces normalized representations.
```
It is unclear how exactly the authors use the linear layer to classify pairs of images.
2. Furthermore, it is not clear why predicting the index of the pair within the given batch results in contrastive loss. Indeed, the model will be penalized for predicting different labels (for the original and augmented images), however, it will also be penalized for predicting the same label for both images but different from their index in the batch. This design choice does not have any motivation in the paper.
3. The variational distribution for the contrastive head layer does not correspond to the probabilistic model used in eq. (3). Indeed, the parameters of the variational family described in Section 3.2 depend on the input data, while, in eq. (3), the authors use it as a conventional variational family that does not depend on inputs.

I also have some minor concerns regarding the presentation:
- throughout the paper, the authors use the word "predictive" as a noun, which I can interpret as "predictive distribution", but it adds confusion;
- it is not clear what "class template vector" means;
- the paper has grammar mistakes, for instance, in Section 3.1, see the second sentence of the first paragraph;
- the paper has spelling mistakes, for instance, in Section 3.1, see the last sentence of the first paragraph.

The empirical evaluation of the proposed approach can be significantly improved.
1. The paper does not contain an ablation study for the introduced design choices. For instance, the choice of the variational distribution and the contrastive loss (which is not properly discussed) are not motivated and are not studied properly.
2. The proposed method is compared only against the methods that do not use the unlabelled data. The comparison against other semi-supervised techniques (not necessarily Bayesian) is required.
3. In Table 5, reported accuracies are much lower than other conventional and Bayesian models. For instance, see Table 9 in [2]. This raises a major concern regarding the quality of the baselines.

[2] Maddox, Wesley J., Pavel Izmailov, Timur Garipov, Dmitry P. Vetrov, and Andrew Gordon Wilson. "A simple baseline for bayesian uncertainty in deep learning." Advances in neural information processing systems 32 (2019).

---

> ### Author Response · Authors · 2024-04-12
> **Response 1/2**
>
> We would like to thank you for your helpful feedback and questions, which have really helped us improve our paper. We will address your concerns in the following. Note that revisions in our updated manuscript are highlighted in green.
>
> > The contrastive loss is not specified in the paper [...] It is unclear how exactly the authors use the linear layer to classify pairs of images.
>
>
> Apologies for not making this clearer in the paper. We use the parameters $\theta^s$ to map our inputs $x$ to some representations $z = f(x; \theta^s)$ and then aim to classify them in the contrastive task using a linear layer with weights $W^c$ as $p(y | x) = \textrm{softmax}(W^c z)$, where the label $y$ is aiming to match the correct index from the batch.
> Crucially, we don't learn $W^c$ directly, but indirectly parameterize our variational distribution $q(W^c)$ through the $z$s, which depend on the the shared parameters $\theta^s$. That is because ultimately, we care about learning a $\theta^s$ that maps the data into a useful representation space. We set the variational distribution for the linear layer as $q(W^c; \tau, \sigma^2) = \mathcal{N}(\mu^c, \sigma^2 I)$, where $\mu^c = [\omega_1^\top \dots \omega_M^\top]/\tau$ with $\omega_i = (z_i^A + z_i^B)/2$. So, in words, the $\omega_i$ are the means of the representations of the same image under two different augmentations, which are the optimal class template vectors to represent class $i$ (i.e., the source image $i$). So the softmax logit for each $z$ being an instance of source image $i$ will be $\omega_i^\top z$.
> Now, the actual loss we optimize, as shown in Algorithm 1, is
>
> $\mathcal{L}(\theta^s, \tau, \sigma) = \log p(\theta^s) + \frac{1}{2M} \mathbb{E}_{q(W^c)} [\log p(\mathcal{D}^c | \theta^s, W^c)] $
>
> $- \bar{D}_{KL} (q(W^c) \| p(W^c)) $
>
> The first term encourages our shared parameters to be close to some weight space prior (in the case of a Gaussian prior, this is equivalent to weight decay). The second term encourages our learnt representations to fit the constrastive task well. This is where the actual contrastive learning happens, namely, the Categorical likelihood yields a cross-entropy loss that encourages the logits $\omega_i^\top z_i$ to be large, that is, the representation $z_i^A$ and $z_i^B$ to be close to their respective mean (and thus, each other). Moreover, it encourages the logits $\omega_i^\top z_j$ to be small for $j \neq i$, that is, it pushes representations $z_i$ and $z_j$ for different source images away from each other. Finally, the last term in the loss encourages $W^c$ to be close to its prior. In the case of a zero-centered Gaussian, this is just a penalty on the norm of the representations $z$ and makes sure that they don't get pushed infinitely far away from each other by the contrastive likelihood term.
> We hope that this clarifies our loss and have added these discussions to the manuscript.
>
>
> > it is not clear why predicting the index of the pair within the given batch results in contrastive loss. Indeed, the model will be penalized for predicting different labels (for the original and augmented images), however, it will also be penalized for predicting the same label for both images but different from their index in the batch.
>
> We hope that the explanation above also clarifies this point. Of course, we agree that the ordering of the source images in the batch, and thus the actual labels for each image are arbitrary, but this is remedied by the fact that $W^c$ is always defined as the mean of the actual representations $z$, so the gradients on $\theta^s$ will be invariant to the ordering of the batch.
>
>
> > The variational distribution for the contrastive head layer does not correspond to the probabilistic model used in eq. (3). Indeed, the parameters of the variational family described in Section 3.2 depend on the input data, while, in eq. (3), the authors use it as a conventional variational family that does not depend on inputs.
>
> We hope that the explanation above also clarifies this. Note that, while Eq. (3) allows for any variational distribution in principle, we describe in the paper just below the equation that we will use the construct explained above. This has many benefits, including the invariance to batch ordering, as pointed out in your previous remark. Also recall that we do not actually use $q(W^c)$ anymore at the prediction time of the model, it is only a tool to help us learn $\theta^s$ and gets discarded after the contrastive learning step.

---

> > ### Author Response · Authors · 2024-04-12
> > **Response 2/2**
> >
> > > throughout the paper, the authors use the word "predictive" as a noun, which I can interpret as "predictive distribution", but it adds confusion
> >
> > Apologies for that, we have clarified that in the manuscript.
> >
> > > it is not clear what "class template vector" means
> >
> > As described above, we mean by that the vector which yields the class logit when taking an inner product with the representation $z$. We have clarified this in the manuscript.
> >
> > > the paper has grammar/spelling mistakes
> >
> > Apologies, we have fixed those.
> >
> >
> > > The paper does not contain an ablation study for the introduced design choices. For instance, the choice of the variational distribution and the contrastive loss (which is not properly discussed) are not motivated and are not studied properly.
> >
> >
> > Apologies again for not motivating the loss in detail. After having added our explanation above, we hope that it has become clear now that this is the only sensible way to yield a contrastive variational objective on the representations $z$, while still satisfying all our desiderata, such as invariance to batch ordering. Additionally, we present an ablation study in Table 4 in the appendix, which shows that a naive variational distribution independent of the representations does indeed not perform well.
> >
> >
> > > The comparison against other semi-supervised techniques (not necessarily Bayesian) is required.
> >
> >
> > We agree. In Figure 5 in the appendix, we present a comparison against SimCLR, a state-of-the-art non-Bayesian contrastive learning technique. We see that our SBNN performs on par with SimCLR in terms of accuracy, while our SBNN* performs slightly better. Moreover, both versions of our approach perform better in terms of uncertainty calibration. We also compare to SimCLR in Figure 4 in the main text, where we also see that our approach yields better uncertainties that are more useful for downstream active learning.
> > Note that the main goal of our work is to show how BNN priors can be improved with self-supervised learning, not necessarily to develop a new state-of-the-art self-supervised learning algorithm. Hence, we believe that an empirical comparison to a wider range of non-Bayesian approaches beyond SimCLR would be out of scope for our work.
> >
> >
> > > In Table 5, reported accuracies are much lower than other conventional and Bayesian models.
> >
> > Our accuracy on the full CIFAR-10 dataset is around 92 %, which is very much in line with state-of-the-art approaches in Bayesian deep learning, such as linearized Laplace inference [1], global inducing point VI [2], SG-HMC [3], and functional VI [4]. In fact, we slightly outperform some of these approaches.
> > Note that we have not spent excessive efforts to tune hyperparameters, which is why we don't reach the performance levels of some highly optimized models from the computer vision community. We believe that the scope of our paper is to show that contrastive learning on unlabeled data can lead to better BNN priors and not to achieve absolute SOTA on computer vision tasks.
> >
> > [1] https://proceedings.mlr.press/v130/immer21a/immer21a.pdf
> >
> > [2] https://proceedings.mlr.press/v139/ober21a/ober21a.pdf
> >
> > [3] https://proceedings.mlr.press/v139/izmailov21a/izmailov21a.pdf
> >
> > [4] https://proceedings.neurips.cc/paper_files/paper/2022/file/8ea50bf458f6070548b11babbe0bf89b-Paper-Conference.pdf
> >
> >
> >
> > We would like to thank you again for your comments and hope to have addressed all your concerns. Please let us know if you have any further questions. Moreover, in the spirit of TMLR, we are open to modifying our claims to reflect our presented evidence better, if you believe that that would be necessary.

---

### Review · Reviewer_84pJ · 2024-03-15

**Summary Of Contributions:**

The work presents advances in Bayesian neural networks (BNNs) leveraged by unlabelled data. For this task, a self-supervised framework is designed around the prior predictive distribution for improving on the NN predictive tasks. Results show that the priors learned by leveraging unlabelled data work better than conventional BNN priors.

**Audience:**

Yes

**Claims And Evidence:**

No

**Requested Changes:**

I want to remark that I have a positive impression of the self-supervised and contrastive learning approach proposed. However, I still feel that it does not fit very well in the current focus and framework of BNNs, to the point, that it makes me doubt if the method is actually a Bayesian technique. So, having some answers and improvements on what I wrote before and sections 3.1 and 3.2 would be a plus. Including some extra clarity in the presentation of the methodology.

**Strengths And Weaknesses:**

#### **Strenghts.**

- the idea and exploitation of unlabelled data for obtaining better priors in BNNs is definitely powerful and interesting.
- authors did a good review of the state-of-the-art, and I find the related work and references particularly well written.
- even if i have concerns with certain points, I did not find critical mistakes in the equations and derivations, and notation fits with the ideas and methods proposed.
- experiments are sufficient, with a decent comparison wrt other SOTA methods from BNN literature.
- the paper has still some issues with clarity, in my opinion, but it is general readable and easy to follow.

#### **Weaknesses.**

I would like to focus my concerns on the concept that the paper brings a lot: "BNN prior predictive / prior predictive distribution". From a Bayesian perspective, this prior predictive is kind of odd to me, because it basically ignores data (there is no inference there), and one samples weights/parameters from a "tuned" prior for later averaging standard likelihood predictions over the test data $y_\star$ (I'm looking to Eq. (2) for instance). This point confuses me a lot because some predictive properties/performance are claimed all along the paper bc this prior predictive distribution is well obtained.

Having some experience in BNNs, I've not seen this prior predictive thing a lot, even if in many cases one wants to do model selection (i.e. fitting hyperparameters of the prior wrt to marginal likelihood) or just posterior approximations for later performing well with the (posterior) predictive estimates.

Before Eq. 2, it is claimed that learning in this way is known as model learning, and used in Wilson et al. 2015, 2016. However, I've not found in these references something similar to the prior predictive framework proposed by the authors, so I have important doubts around the validity of such claims.

I feel that the introduction of self-supervised learning/ contrastive methods with unlabelled data for improving BNNs is definitely a big topic to come, but from section 3.1 and partially section 3.2, I feel that some Bayesian principles are not entirely respected or well adjusted to what authors would like to propose in the paper.

---

> ### Author Response · Authors · 2024-04-12
> **Response**
>
> We would like to thank you for your helpful feedback and questions, which we will address in the following. Note that revisions in our updated manuscript are highlighted in green.
> We are encouraged to see that you find our idea powerful and interesting, our experiments convincing, and our paper well-written.
>
>
> > I would like to focus my concerns on the concept that the paper brings a lot: "BNN prior predictive / prior predictive distribution". From a Bayesian perspective, this prior predictive is kind of odd to me, because it basically ignores data (there is no inference there)
>
> Our prior predictive $\mathbb{E}_{p(\theta)}[p(y | x, \theta)]$ is indeed not often considered in standard treatments of Bayesian deep learning, but it is in fact closely related to function-space priors $p(f)$.
> In traditional Bayesian function-space methods, such as Gaussian processes (GPs), the prior is often characterized by its conditional distributions at different inputs $x$ (which in case of a GP are all Gaussian).
> In BNNs, we cannot easily write down these distributions in closed form, but we can still study them empirically and demand that they have sensible properties. In our case, we demand that in expectation, these function-space priors make predictions that are reasonably classifying the data on some semantically relevant task, hence we study the prior predictive on contrastive tasks. We have clarified this in the paper.
>
> > it is claimed that learning in this way is known as model learning, and used in Wilson et al. 2015, 2016. However, I've not found in these references something similar to the prior predictive framework proposed by the authors
>
> In these references, the marginal likelihood is used to select function-space priors (in the form of GPs) that explain (some version of) the data well. Crucially, these priors are parameterized by a point estimate of the kernel parameters. In our case, we also use a lower bound on the marginal likelihood (i.e., the ELBO) to select function-space priors (as described above), parameterized by the shared parameters in the lower layer of the BNN, that explain the contrastive data well. These shared parameters can even be viewed as a kernel, the RKHS of which is given by the representation space that these layers map the inputs into. While this is not exactly the same approach, we believe that some of the intuition should transfer. See also the last paragraph of Section 3. We have also added a more in-depth discussion of this to the paper.
>
> > I feel that the introduction of self-supervised learning/ contrastive methods with unlabelled data for improving BNNs is definitely a big topic to come, but from section 3.1 and partially section 3.2, I feel that some Bayesian principles are not entirely respected
>
> We want to stress that our graphical model in Figure 2 is still a fully proper Bayesian model, adhering to all the principles. Since we are working with neural networks, we can naturally not resort to exact inference, so we have to make the standard approximations that are common in the field, such as variational inference, Laplace inference, and partial stochasticity. We would argue that overall this is not less Bayesian than other Bayesian deep learning papers.
>
>
> > Including some extra clarity in the presentation of the methodology.
>
> Thanks, we have made adjustments to the presentation.
>
> We would like to thank you again for your comments and hope to have addressed all your concerns. Please let us know if you have any further questions. Moreover, in the spirit of TMLR, we are open to modifying our claims to reflect our presented evidence better, if you believe that that would be necessary.

---

### Decision · Action_Editor_9M7S · 2024-05-15

**Recommendation:** Accept with minor revision

**Comment:**

As discussed above, the paper is currently primarily lacking in its empirical evaluation, which needs to be rectified. The aim of this revision should not be to chase state-of-the-art results in the fully supervised case, i.e., 50000 labels, but to ensure that the reported results for each setup mirror their prior reported performances for the reader to accurately interpret the results.

I recommend this as a minor rather than as a major revision, as I expect the relative performance improvements to persist in the more relevant semi-supervised setups.
While not mandatory for acceptance this would also allow the authors to provide the reader with error bars on all reported results, which would further improve the interpretability of the results.

I therefore recommend _Acceptance with minor revision_.

**Audience:**

Improving the performance of BNNs and training with unlabelled data, in general, are still active areas of research within the TMLR community. As such, I expect the current work to be of interest to TMLR's audience.

**Claims And Evidence:**

The authors propose a new way to incorporate unlabelled data to improve the training and performance of BNNs. While the reviewers agree that the task is an important one and the solution is a new one four main areas of weaknesses were highlighted. (i) The Bayesianity of the method; (ii) the contrastive formulation and presentation; (iii) its empirical evaluation; (iv) interpretation and relation to prior work.

Most of these points have been addressed in the revision of the paper, with the main remaining one being its empirical part. While it is explicitly not the case that the merit of a TMLR submission should be judged on whether it achieves state-of-the-art results, it is concerning if the results reported for prior work are worse than those reported in the literature.

See, e.g., reviewer R7PH's Maddox et al. (2019) reference and the results therein, as well as, Rudner et al. (2022) (see ref [4] in the rebuttal response), Table 2 vs Table 5 in the submission. In both cases, the baselines report a lot higher performance numbers than are reported in this submission.

Before publication, the authors need to ensure that the improved performance of their method relative to prior work is not due to suboptimal training of those baselines.